# Leveraging place field repetition to understand positional versus nonpositional inputs to hippocampal field CA1

**William Hockeimer**[1,2]*†, **Ruo-Yah Lai**[1], **Maanasa Natrajan**[2], **William Snider**[1,2], **James J Knierim**[1,2,3]*

[1]Krieger Mind/Brain Institute, Johns Hopkins University, Baltimore, United States; [2]Solomon H. Snyder Department of Neuroscience, Johns Hopkins University School of Medicine, Baltimore, United States; [3]Kavli Neuroscience Discovery Institute, Johns Hopkins University, Baltimore, United States

**Abstract** The hippocampus is believed to encode episodic memory by binding information about the content of experience within a spatiotemporal framework encoding the location and temporal context of that experience. Previous work implies a distinction between positional inputs to the hippocampus from upstream brain regions that provide information about an animal's location and nonpositional inputs which provide information about the content of experience, both sensory and navigational. Here, we leverage the phenomenon of 'place field repetition' to better understand the functional dissociation between positional and nonpositional information encoded in CA1. Rats navigated freely on a novel maze consisting of linear segments arranged in a rectilinear, city-block configuration, which combined elements of open-field foraging and linear-track tasks. Unlike typical results in open-field foraging, place fields were directionally tuned on the maze, even though the animal's behavior was not constrained to extended, one-dimensional (1D) trajectories. Repeating fields from the same cell tended to have the same directional preference when the fields were aligned along a linear corridor of the maze, but they showed uncorrelated directional preferences when they were unaligned across different corridors. Lastly, individual fields displayed complex time dynamics which resulted in the population activity changing gradually over the course of minutes. These temporal dynamics were evident across repeating fields of the same cell. These results demonstrate that the positional inputs that drive a cell to fire in similar locations across the maze can be behaviorally and temporally dissociated from the nonpositional inputs that alter the firing rates of the cell within its place fields, offering a potential mechanism to increase the flexibility of the system to encode episodic variables within a spatiotemporal framework provided by place cells.

*For correspondence: wih38@pitt.edu (WH); jknierim@jhu.edu (JJK)

Present address: †Rehab Neural Engineering Labs, University of Pittsburgh, Pittsburgh, United States

Competing interest: The authors declare that no competing interests exist.

## Editor's evaluation

This is a fundamental work that convincingly reveals that place cells in the hippocampus that exhibit repeated firing fields incorporate information about non-positional variables in each firing field. The authors reveal that individual firing fields of a single place cell can exhibit tuning to different head orientations, suggesting hippocampal neurons are flexible in terms of how they incorporate non-positional inputs.

## Introduction

All behavior is embedded within a subjective spatial and temporal framework. In the mammalian brain, allocentric space is represented within a distributed, interconnected network of regions that specialize in different aspects of spatial experience (*O'Keefe, 1976*; *Taube, 1998*; *Hafting et al., 2005*; *Alexander and Nitz, 2017*; *Nitz, 2012*; *Sharp, 1999*). This navigation system is critically dependent on the hippocampus, whose pyramidal cells have spatial firing fields that are distributed throughout the environment to provide a continuous representation of position (*O'Keefe and Dostrovsky, 1971*; *O'Keefe, 1976*). The hippocampus both receives and generates spatial signals as part of an entorhinal-hippocampal-subicular processing loop (*Witter and Amaral, 2004*). The CA1 region receives spatially tuned inputs from regions such as CA3 (other place cells) and the entorhinal cortex (grid cells, spatial non-grid cells, landmark-vector cells) that influence where a CA1 place field is located. These place cells are thought to form the basis of a cognitive map within which the 'items and events of an organism's experience are located and interrelated' (*O'Keefe and Nadel, 1978*, p. 1) in support of episodic memory.

The hippocampus also receives direct and indirect inputs from regions not involved in position signaling, such as high-level association areas, head direction cells in the anterior dorsal nucleus of the thalamus, and the amygdala, which provide information about the local or internal environment. Neurons within the hippocampus respond to a wide array of sensory and otherwise nonspatial cues, with CA1 being the most studied region. CA1 neurons respond to odors (*Wiener et al., 1989*; *Wood et al., 1999*; *Wiebe and Stäubli, 1999*; *Komorowski et al., 2009*; *Allen et al., 2016*), tones (*Moita et al., 2003*; *Aronov et al., 2017*), textures (*Itskov et al., 2011*), and tastants (*Herzog et al., 2019*; *Herzog et al., 2020*), and they can be influenced by internal variables such as subjective frames of reference (*Gothard et al., 1996*; *Jackson and Redish, 2007*; *Fenton et al., 2010*). According to the cognitive map theory (*O'Keefe and Nadel, 1978*), these variables are represented in the spatial place map to incorporate details about the individual components of an experience and bind them together to form episodic memory traces. Hippocampal neurons also respond to variables that are navigational but nonpositional. For example, neurons have been recorded that respond to travel direction (*McNaughton et al., 1983*; *Wiener et al., 1989*; *Markus et al., 1995*), prospective and retrospective turns or positions (*Wood et al., 2000*; *Ferbinteanu and Shapiro, 2003*; *Smith and Mizumori, 2006*; *Ji and Wilson, 2008*), entire trajectories (*Grieves et al., 2016a*), and speed (*McNaughton et al., 1983*; *Czurkó et al., 1999*; *Maurer et al., 2012*; *Huxter et al., 2003*; *Wiener et al., 1989*). These variables relate to navigation, but do not explicitly encode the current position.

In many of the examples above, and related studies, the firing rates of the neurons changed while the locations of the firing field stayed stable (*Leutgeb et al., 2005*; *Herzog et al., 2019*; *Moita et al., 2003*; *Wood et al., 2000*). It should be noted that in at least one study some place fields shifted their center of mass in response to combinatorial cue changes, although this study also reported in-field firing rate changes as well (*Anderson and Jeffery, 2003*). These findings, termed 'rate remapping', inspired the hypothesis (*Leutgeb et al., 2005*) of a functional dissociation between where the neuron fires and the quantitative rate of firing within the field (see also *Shapiro et al., 1997*; *Manns and Eichenbaum, 2009*). According to this hypothesis, the location of the field is used to encode position while the exact firing rate varies as a function of nonpositional cues (so long as the firing rate is maintained above some threshold to indicate the field is active), thereby allowing the formation of distinct episodic memories within a stable spatial framework. The common representation, via rate remapping, of both sensory cues and navigational cues unrelated to position suggests that they are both categorized together as nonpositional variables distinct from the positional signal that is the primary behavioral correlate of hippocampal firing in rodents.

It is poorly understood how positional and nonpositional signals are integrated within individual place cells. Place cells with multiple fields in an environment provide an opportunity to infer the functional difference between these signals by allowing a within-cell comparison of multiple nonpositional inputs across multiple place field locations. CA1 neurons have multiple fields under different circumstances, for instance in large environments (*Rich et al., 2014*; *Lee et al., 2020*; *Park et al., 2011*; *Harland et al., 2021*) or environments with repeated structural elements. The latter, termed 'place field repetition' (*Spiers et al., 2015*; *Grieves et al., 2017*), is defined as a neuron with multiple spatial firing fields at instances of a repeated structural element or behavioral motif, such as a repeated trajectory, in an environment. For CA1, many different types of repeating track structure can elicit

place field repetition, e.g., multiple boxes arranged in a row (*Skaggs and McNaughton, 1998*; *Spiers et al., 2015*; *Grieves et al., 2016b*), a spiral track (*Cowen and Nitz, 2014*), a spiral staircase (*Hayman et al., 2011*), or an M-maze or hairpin maze (*Frank et al., 2000*; *Derdikman et al., 2009*; *Singer et al., 2010*). Repetition has also been observed outside of the hippocampus, in the entorhinal cortex (*Frank et al., 2000*), subiculum (*Stewart et al., 2014*), and retrosplenial cortex (*Alexander and Nitz, 2017*). Here, we show that repeating fields of the same neuron do not always display the same nonpositional rate modulation, demonstrating that nonpositional cues are dissociable from the positional inputs onto place cells in a given environment.

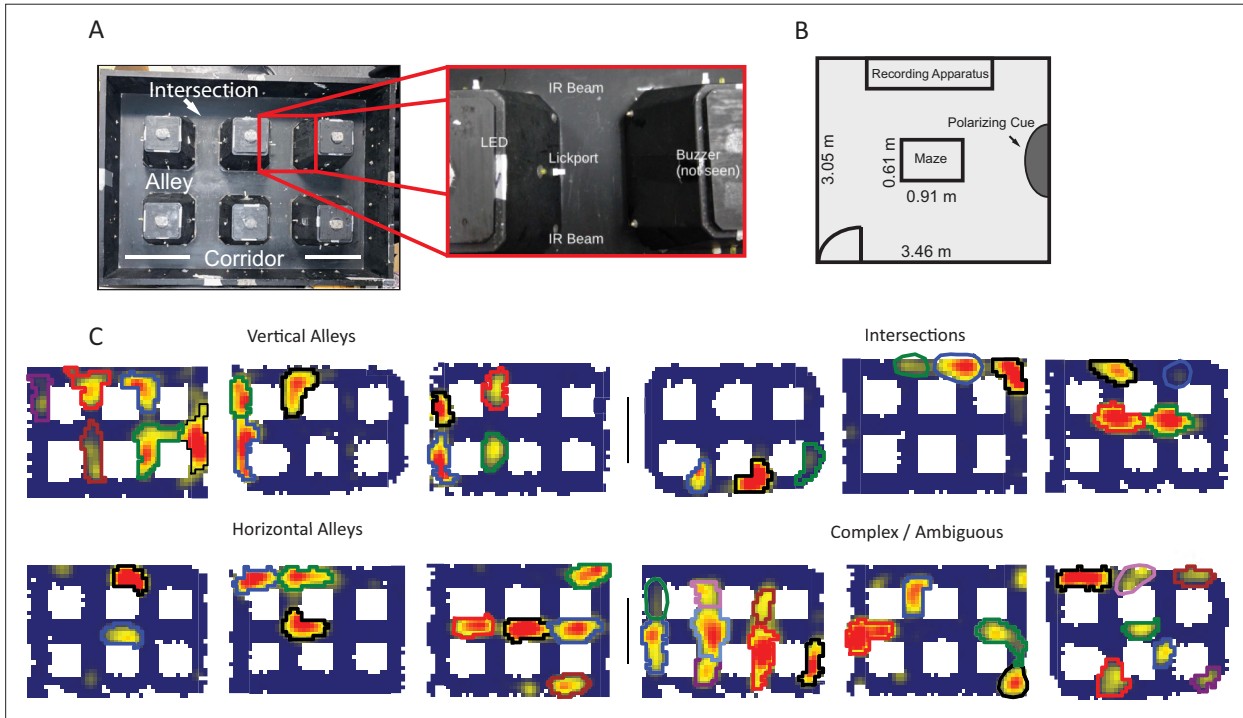

**Figure 1.** Apparatus and examples of place field repetition. (**A**) Left, top-down view of city-block maze. Perimeter walls (15.2 cm high) enclose an area 60.96×91.44 cm². Six square obstacles (19.1×19.1 cm², 15.2 cm tall) define 17 alleys in their interstitial spaces along with 12 intersections. Track regions were defined as vertical alleys (oriented north-south), horizontal alleys (oriented east-west), or intersections. Right, enhancement showing an individual alley. An IR emitter and detector pair was affixed to either entrance of the alley. In the center of the alley was a lickport with a visible light LED above. A buzzer was placed under each alley. Lickports here have been extended to make them more visible. (**B**) Schematic of recording room and behavioral apparatus. The city-block maze was in the same room as the recording apparatus. A black, floor-to-wall curtain that was gathered together, forming a band along the east wall, served as a polarizing cue. (**C**) Examples of repeating fields grouped into four subjective categories: repeating in vertical alleys, repeating in horizontal alleys, repeating in intersections, and complicated or ambiguous patterns of repetition. Spatial bins from off-track sampling were removed from these figures and from quantitative analyses of the place fields. Each identified subfield is outlined in a separate color.

The online version of this article includes the following figure supplement(s) for figure 1:

**Figure supplement 1.** Histology from representative animals.

**Figure supplement 2.** Behavioral sampling.

**Figure supplement 3.** All ratemaps for R765.

**Figure supplement 4.** All ratemaps for R781.

**Figure supplement 5.** All ratemaps for R808.

**Figure supplement 6.** All ratemaps for R859.

**Figure supplement 7.** All ratemaps for R886.

**Figure supplement 8.** Behavioral processing pipeline.

**Figure supplement 9.** Putative interneurons.

# Results

Hippocampal CA1 cells (*Figure 1—figure supplement 1*) were recorded as rats navigated a novel city-block maze for pseudorandomly delivered reward (*Figure 1—figure supplement 2*). Although the task shares similarities to open-field foraging tasks in that the animal freely navigated for pseudo-random reward, the behavior was not random, as the animals displayed variable degrees of stereotypy in their trajectories (*Figure 1—figure supplement 2*). The track consisted of 17 interlocking alleys and 12 intersections between them (*Figure 1A and B*). This design allowed us to test whether individual place fields would show repeating firing fields at geometrically similar locations and, if so, whether the repeating place fields would show identical directionality properties (a well-studied example of a nonpositional firing correlate of place fields in one-dimensional [1D] tasks). Each recording day consisted of a single behavioral session, typically 1 hr in duration. A baseline period, in which the rat was confined to a small holding dish next to the maze, was recorded immediately before and after the experimental session for at least 15 min each. On average, 14.7 cells per session (5–35 cells) were recorded that met the inclusion criteria for detailed analysis (see Materials and methods).

## Place field repetition

Neurons with multiple fields at similar types of locations, termed 'repeating place cells' (*Spiers et al., 2015*), were observed (*Figure 1C*). Fields repeated at alleys and intersections within the maze (*Figure 1A*). A repeating place cell was defined operationally as any neuron with two or more place fields in a common type of location: vertical alley, horizontal alley, or intersection. This criterion did

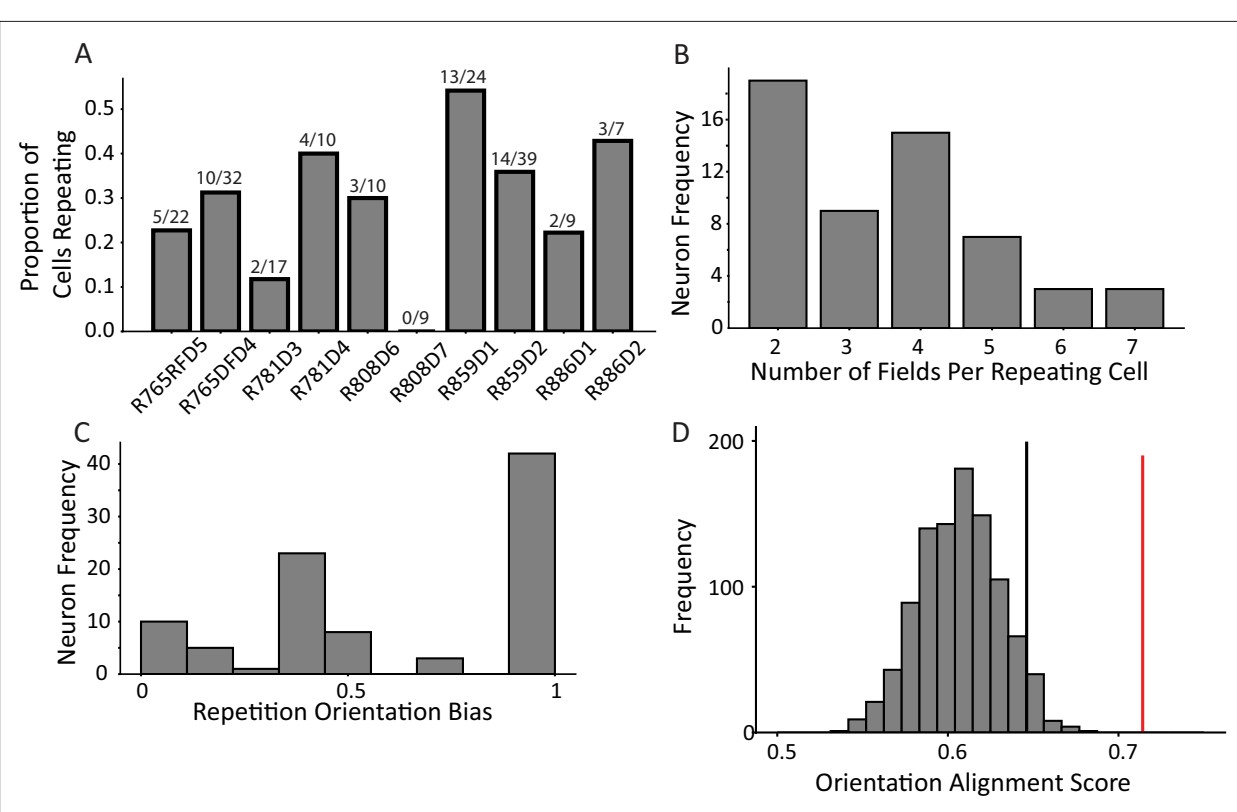

**Figure 2.** Characterization of repetition. (**A**) The frequency of place field repetition across days and rats as defined operationally (two or more fields in a common location). Each bar shows the proportion of well-isolated cells with on-track firing that was repeating, from ten datasets across five rats. (**B**) The distribution of number of fields within repeating cells. Repeating cells had between two and seven detected fields, with half of cells displaying four or more fields. (**C**) The orientation bias among repeating fields in alleys. Cells were assigned an 'orientation bias score' by taking the ratio between the maximum number of fields sharing a common orientation and the total number of fields. (**D**) To quantify repetition, the orientation alignment score (OAS) was defined. The metric first defines the degree of alignment of the fields of a cell, and then expresses that as a percent along the range of possible alignments for that given the number of fields the cell had (see Materials and methods). The y-axis indicates frequency of shuffles of the entire population that had the average OAS indicated on the x-axis. The mean of the population 0.71 (red) was greater than the 95th percentile of a shuffle distribution (0.64, black).

not penalize cells with violations of a repeating pattern, for instance two fields in vertical alleys and one in a horizontal alley. This was justified by the complex nature of the repeating phenomenon observed (*Figure 1—figure supplements 3–7*). Repeating fields were equally likely to be in vertical or horizontal alleys (167 horizontal versus 146 vertical fields, chi-square goodness of fit, compared to a null distribution of equal numbers of horizontal and vertical fields, $\chi^2(1)=1.41$, p=0.235). Each repeating cell had up to seven fields, suggesting that repetition was spatially modulated at the level of the apparatus, as no cell had fields at each alley (mean number of repeating fields = 3.55 ± 1.46 fields; *Figure 2B*). Many cells had fields that overlapped multiple regions, e.g., fields which curled around the corner of a block (71.89% of fields in one alley; 22.7% of fields in two alleys; 5.1% of fields in three alleys; 0.4% of fields in four alleys). These cases were assigned to the more general three categories (vertical alley, horizontal alley, and intersection) by splitting fields based on the alleys they overlapped and recombining based on orientation (see Materials and methods). Traditional, single-field place cells were also observed in these data as expected (60/179 units). Place field repetition was not driven by the presence of poorly isolated units that combined multiple fields into one 'cell', as there was no significant difference in cell quality between repeating and non-repeating neurons included in the analysis (Mann-Whitney U=3214, p=0.1).

Place cell repetition was quantified by analyzing the degree to which fields of multi-fielded cells were aligned in alleys of a common orientation. The rationale for this approach was that given the effect of head direction on repetition (*Grieves et al., 2016a*; *Harland et al., 2017*), cells repeating in response to a preferred local pattern of borders (or the paths available around such borders) would manifest as cells with fields aligned in alleys of a common orientation. First, the distribution of field alignment scores was calculated for each multi-fielded neuron as the sum of fields aligned in a common orientation divided by the total number of fields (*Figure 2C*). This distribution was skewed toward 1.0, suggesting that fields tended to share an orientation. An orientation alignment score (OAS) was defined as the percentage of maximal field alignment given the number of fields the cell had (see Materials and methods). The mean OAS of multi-fielded cells was significantly greater than expected by chance (mean = 0.71, 95th percentile shuffle = 0.64, *Figure 2D*). This result demonstrates that multiple fields of a neuron tend to share a positional input that instructs them where to fire. If they did not share such a positional input, their spatial distribution would be random across the track. The fact that the distribution of multiple fields instead respects the orientation of the alleys suggests that there is a common signal that influences their position.

## Directionality of CA1 place fields

An example of a nonpositional influence on place cell firing is directionality. Place fields are known to be directional when rats run on linear tracks or display other types of 1D, stereotyped behavior, but they are much less directional when rats perform more unstructured, 2D behavior in open-field foraging (*Markus et al., 1995*; *McNaughton et al., 1983*; *Navratilova et al., 2012*). Because our city-block maze was a hybrid of these types of tasks (i.e. rats ran in 2D searching for pseudorandom reward like open-field foraging, but their trajectories were constrained to orthogonal directions by the city-maze architecture), we first needed to determine whether place fields were directional in the maze. Continuous, 2D travel through the maze was parameterized according to which alley the rat currently occupied. The current direction was defined based on the overall direction of travel through the alley, as opposed to the instantaneous head direction. For example, if the animal entered a horizontal alley from the west entrance and exited from the east entrance, then the activity during the time spent in that alley would be classified as 'east' regardless of the moment-by-moment head direction (the cardinal directions are defined relative to the camera image, not relative to a compass). First, ratemaps were created and filtered by travel direction to yield four ratemaps based on each cardinal direction (*Figure 3A*). These ratemaps revealed place fields with firing rates that qualitatively varied as a function of direction. For example, for neuron 1 of *Figure 3A*, fields 2 and 3 fired more strongly when the animal traveled from east to west, whereas field 1 shifted toward the direction of travel (i.e. it shifted 'west'). Further, field 2 not only fired more strongly but shifted its center of mass when the animal traveled west. Neuron 2 had multiple firing fields in vertical alleys or intersections. Field 1 fired when the animal was traveling south but was virtually silent when traveling north; field 2 fired more strongly when the animal traveled north than south; and field 3 appeared untuned to travel direction. Neurons 3 and 4 provide similar examples.

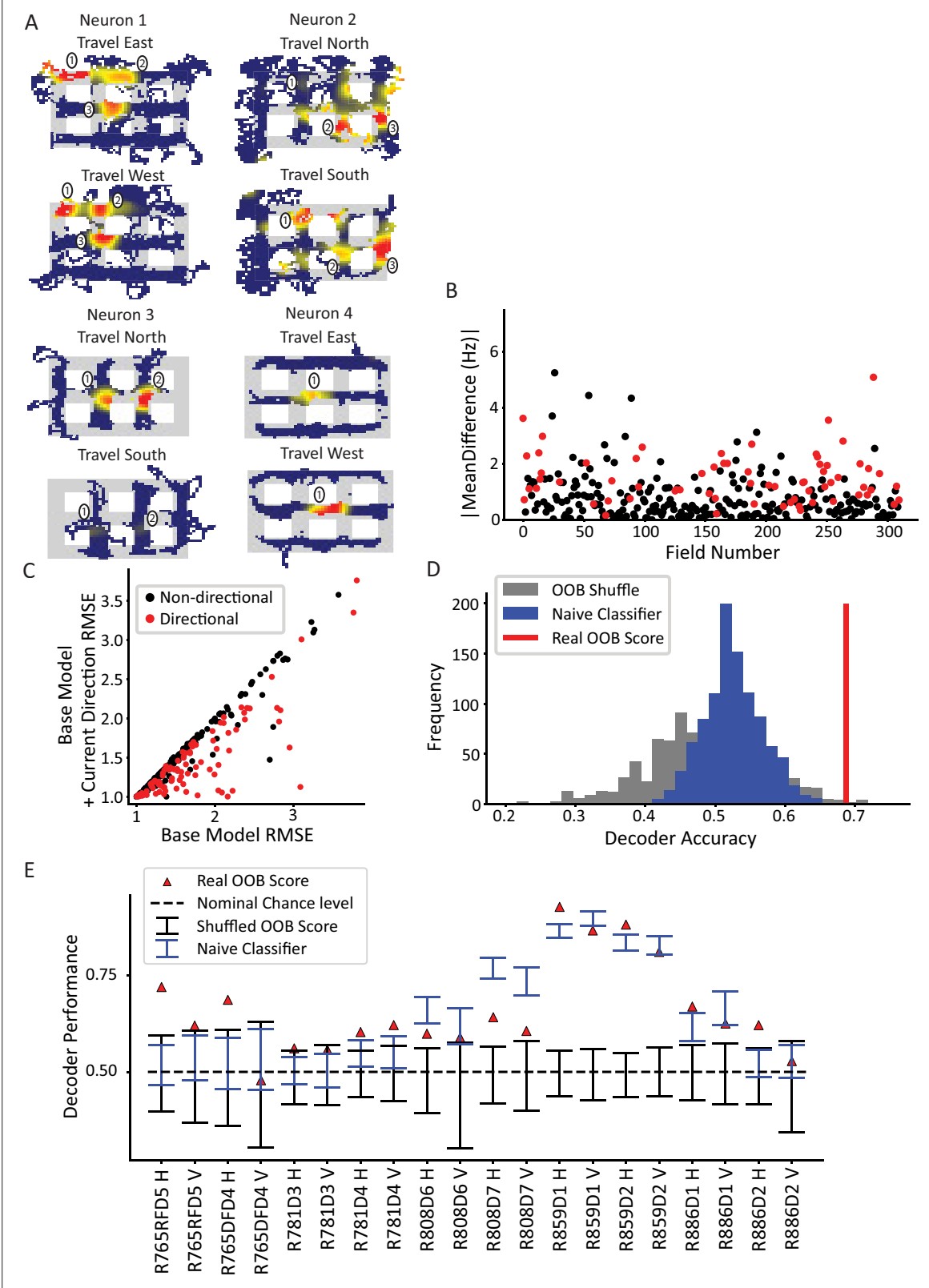

**Figure 3.** Directionality of place fields in the city-block maze. (**A**) Examples of directional place fields. Only unrewarded, straight-through traversals were included for these ratemaps and subsequent analyses. Top left, example neuron with east- and west-oriented alley traversals separated into ratemaps (north and south ratemaps not shown here). Fields 2 and 3 fired more strongly for west over east, while field 1 was untuned. Top right, cell with at least five firing fields in vertical alleys, three of which are labeled. Field 1 preferred south, while field 2 preferred north. Field 3 was untuned. Bottom left, cell

*Figure 3 continued*

with two fields in vertical alleys of adjacent corridors. Both fields fired more strongly when the animal traveled north. Bottom right, cell with single field that fired more strongly when animal traveled west. (**B**) The directionality across population of fields. For each field in an alley with sufficient sampling, passes were orientation-filtered. The absolute mean difference in firing rate was computed between directions. A significant number of fields were directional (93/310 field portions, binomial test, p=3.6 × 10$^{-45}$, FWER-adjusted alpha = 0.05). Two outliers were removed from the visualization (but not the statistics): field 132, normalized Hz = 19.08 (significant) and field 213, normalized Hz = 9.52 (significant). (**C**) Generalized linear models (GLMs) were fit to the orientation-filtered data from each field. Points labeled in red were significant according to a likelihood ratio test (LRT) between the base and directional GLMs. Root mean square error (RMSE) is shown. The current direction model better explained the data in a significant number of cases (103/297 field pieces, binomial test, p=4.53 × 10$^{-57}$, FWER-adjusted alpha = 0.05). One outlier was removed from visualization but not statistics with base RMSE = 4.72, current direction RMSE = 4.67, not significant. (**D**) Example random forest decoder results for a single orientation-filtered dataset. A random forest classifier decoded current direction from the population firing rates on each day of recording separately. Alleys were divided into horizontal and vertical alleys and the traversals within them pooled, yielding two data points for each recording day. Random forests with shuffled labels (black) and naïve classifier (blue) were created and the empirical performance (red) was defined as significant if it exceeded the 95th percentile of both control distributions. (**E**) Random forest classification performance across 10 days of recording, broken down by alley orientation. The classifier was able to consistently extract a directional signal from 10/20 datasets (binomial test, p=1.13 × 10$^{-8}$, FWER-corrected alpha = 0.05).

The online version of this article includes the following figure supplement(s) for figure 3:

**Figure supplement 1.** Permutation shuffling test of current direction generalized linear model (GLM).

**Figure supplement 2.** Lack of encoding of retrospective or prospective direction.

The directional firing of each field was quantified. Fields were filtered such that (a) there were at least two passes in each direction and (b) passes in which the rat was rewarded or did not fully traverse the alley (i.e. the rat turned around) were excluded. The firing rate assigned to each pass for a neuron was computed as the mean of the firing rate over the length of the pass, normalized by the session-averaged ratemap for that alley (i.e. the pass was normalized by its location within the field). This normalization was done to correct for passes that might have elicited different firing rates simply because the rat may have traversed different portions of the field during different passes. This is a particular concern when analyzing directional variables, because the rat may be biased to run a certain way through the alley when performing a specific behavior (e.g. it may run closer to the east wall before making a right turn and closer to the west wall before making a left turn).

Across the population, a significant subset of fields was directionally tuned (***Figure 3B***; 93/310 fields passed a Mann-Whitney test per field; binomial test, p=3.6 × 10$^{-45}$). The mean difference in normalized directional firing was 1.67±2.20 for fields that passed the Mann-Whitney test and 0.69±0.72 for those that did not. In other words, directional fields had a difference in normalized firing that was two-thirds more than their mean firing. The directionality of individual fields was corroborated by a generalized linear model (GLM) analysis. A field may appear directional due to nonspecific changes over time that happen to correlate with biases in behavioral sampling. To account for this, the GLM was first fit with a term for time (base GLM) and then a term for direction was added (alternative GLM). These two models were compared via a likelihood ratio test (LRT) to determine whether the addition of current direction in the model resulted in a significant improvement in the model fit over and above the explanatory power of time. This analysis identified a subset of fields with a significant effect of current direction (***Figure 3C***, 103/297 fields, binomial test p=4.53 × 10$^{-57}$; note that the total number of fields, i.e., the denominator, is different here than for the Mann-Whitney test due to the sampling requirements of the two tests [see Materials and methods]). The GLM results were corroborated with a randomization test which shuffled directional labels before repeating the model comparison process; this procedure resulted in an average of 14.8% fields with a fictive effect of direction, well below the experimental value of 35% (***Figure 3—figure supplement 1***).

Next, the CA1 ensemble was tested for directionality at the population level. A random forest classifier was used to decode the current direction from the population vector of average firing rates (***Figure 3D***). The average firing rate for each neuron was computed for each pass through an alley and labeled with the direction of travel for that pass. Data from vertical and horizontal alleys were classified separately and the out-of-bag (OOB) score (mean = 0.68 ± 0.13) was used as the performance metric. Two controls were used to judge the statistical significance of the performance of the classifier. The first method shuffled the trial labels with respect to the population vectors and the classification was repeated using the (shuffled) OOB score as the performance metric. The second method was a naïve classifier that made a class label prediction in proportion to the behavioral

sampling bias in favor of that directional label. The rationale for the naïve classifier test is that a real classifier could potentially utilize class imbalances (e.g. bias to travel north through a given alley) and consistently guess the majority class to achieve above-chance performance. The naïve classifier mimics this strategy by guessing, for each alley traversal, direction labels in proportion to their behavioral sampling frequency over the entire session. The real performance of the classifier was judged significant for each session if it exceeded the 95th percentile of each test. The random forest classifier successfully decoded direction from the population activity in 10/20 orientation-filtered datasets (binomial test p=1.13 × 10$^{-8}$, FWER-corrected alpha = 0.05, *Figure 3E*). On a day-by-day basis, the classifier performance was variable. This may have been caused by differences in cell yield – some days only contained a few cells, which would lead to poorer spatial coverage of the maze. Relatedly, it is possible that days with poorer behavioral sampling (though still good enough to pass the sampling thresholds) led to poorer decoding as there were fewer samples to decode at under-sampled locations.

Previous studies have shown that CA1 fields respond to future or previous directions along the present trajectory (i.e. noncurrent directions) as well as the current direction, which has been interpreted as encoding the spatial decisions along a learned trajectory (*Ferbinteanu and Shapiro, 2003*; *Smith and Mizumori, 2006*). The presence of retrospective or prospective coding was tested for each field. A GLM was used with regressors for current direction, previous direction, next direction, and, for the reasons mentioned above, time. The number of orientation-filtered fields with a significant effect of prospective or retrospective direction was not more than expected by chance by shuffling directional labels 1000 times and re-running the GLM analysis. However, the effect of retrospective direction trended toward significance (*Figure 3—figure supplement 2*), indicating that there may be a possible weak effect of retrospective direction in the dataset.

## Repeating versus non-repeating fields

It has been hypothesized that repeating neurons have multiple fields because they are tuned to similar paths that are performed at each field location ('path equivalence', *Frank et al., 2000*; *Singer et al., 2010*). If this is the case, then one may expect repeating neurons to be more directional than non-repeating neurons. However, the distributions of average normalized firing rate differences in each direction (analyzed as in *Figure 3B*) were not statistically different between repeating and non-repeating neurons, with a mean difference of 0.88 normalized Hz for the repeating fields and 1.21 normalized Hz for the non-repeating fields (Mann-Whitney U=11,883, p=0.49; *Figure 4A*). To corroborate this lack of difference, a GLM LRT test was used as before (*Figure 4B*). Here, too, there was no difference in the strength of tuning as assessed by differences in the root mean square error (RMSE) between the base model and the current direction model, as the mean improvement in model fit was not significantly different between the groups (Mann-Whitney U=9764, p=0.088). Further, the proportions of directional fields were not different between the groups (69/172 repeating fields versus 51/125 non-repeating fields, Pearson's chi-square $\chi^2$(1)=0.001, p=0.97). These data indicate that repeating fields were equally likely to be directionally tuned as non-repeating fields and those that were directional in each group had similar tuning strength.

The accuracy of position decoding from populations of repeating place cells remains unclear given that their discharge is spatially ambiguous, yet fewer cells are needed to cover a given environment. A linear regression classifier was used to decode position from a population vector of neural firing rates. Spikes were convolved to yield a firing probability trace over time for each neuron. At each timepoint, the vector of the population activity of all cells (repeating and non-repeating) was used to predict the x and y coordinates of the position, separately. The coefficient of determination (r$^2$) was used as the readout of decoder performance. Position could be reliably decoded from the population activity (median performance above 95th percentile of shuffled data in 10/10 datasets, binomial test p=9.76 × 10$^{-14}$; *Figure 4C*). These data also undercut the hypothesis that repetition simply reflects confusion within the network given the ambiguous environment, since spatial decoding is still robust and a substantial number of neurons do not repeat (*Figure 2*) (and see *Singer et al., 2010*). Repeating and non-repeating cells were then analyzed separately. The difference in decoding performance between subpopulations of repeating and non-repeating neurons (with the larger group downsampled) shows that both groups of neurons were similarly informative about position (average difference = 0.06 ± 0.21, t-test compared to 0, t(8) = 0.923, p=0.38; *Figure 4D*).

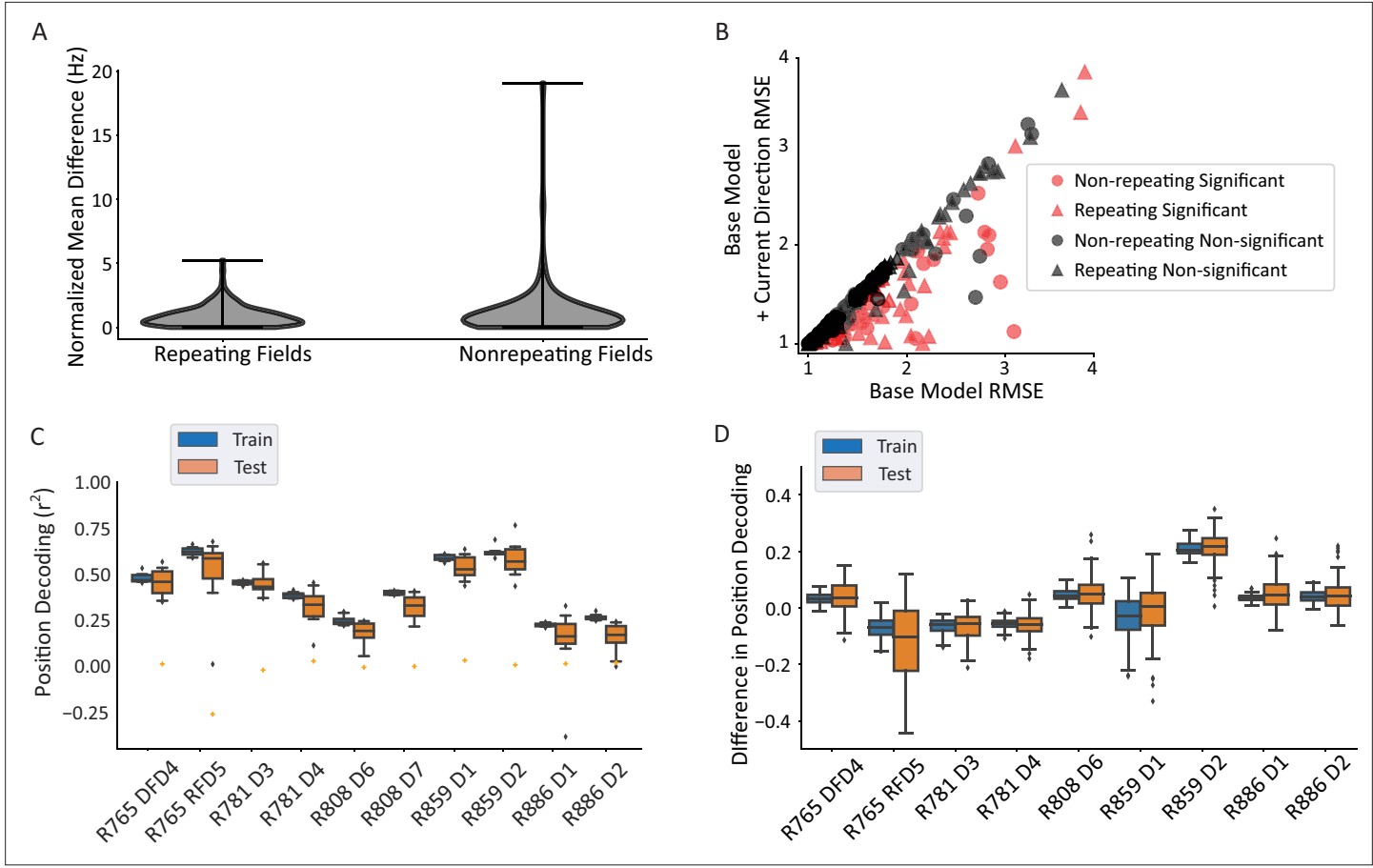

**Figure 4.** Repeating versus non-repeating fields directionality and spatial decoding. (**A**) The distribution of directional firing metrics for each repeating or non-repeating field. Directional firing was computed, as above, as the unsigned difference between the average firing rate in each of the two directions through the field. The two populations were not different from one another (repeating fields 0.88±0.81 versus non-repeating fields 1.21±1.97 normalized Hz; Mann-Whitney U=11,883, p=0.49). (**B**) Generalized linear model (GLM) model comparison as in *Figure 3C*, separated by whether the field was repeating or not. There was no difference in either the proportion of fields that were significant in each category (69/172 repeating field portions versus 51/125 field portions, $\chi^2$(1)=0.001, p=0.97) or the degree of improvement to the model fit (root mean square error [RMSE] differences in base versus direction model, repeating=–0.104, non-repeating=–0.145, Mann-Whitney U=9764, p=0.088). The same outlier was removed from visualization as in *Figure 3C*. (**C**) Position decoding from training and testing data across datasets. Each box plot is the decoding performance (from training or test epochs) from 1000 rotations of the training/test window boundaries. Orange dot indicates the 95th percentile of a shuffled distribution formed by circularly shuffling the rat's position with respect to the neural population activity vector. The median decoder performance was 0.38±0.29 (IQR). (**D**) The difference in decoder performance from datasets in C using only repeating or non-repeating neurons. (See *Figure 4—figure supplement 1* for spatial decoding by repeating or non-repeating cell type.) The group with the smaller sample size was downsampled 10 times per rotation of the train/test windows and the performance averaged. Non-repeating performance was subtracted from repeating performance and the distribution of differences is shown for train and test sets. Most datasets had differences near 0, indicating that overall, there was little difference in decoding performance for repeating versus non-repeating cells.

The online version of this article includes the following figure supplement(s) for figure 4:

**Figure supplement 1.** Continuous decoding by repeating type.

## Nonconservation of nonpositional inputs across repeating fields

Place field repetition offers an opportunity to better understand how CA1 place cells integrate the positional inputs that determine their firing locations with the nonpositional inputs that modulate the firing rates in their fields. The OAS analysis in *Figure 2D* demonstrates that many neurons with multiple fields share a common input instructing them about what type of position to fire in – vertical or horizontal alleys. Because repeating fields share a common positional signal, they can serve as an internal control to test independently the contributions of the nonpositional signals. One possibility is that repeating fields share their directional tuning, i.e., both the positional and nonpositional properties of a place field are repeated across the environment. Indeed, as stated above, it has been

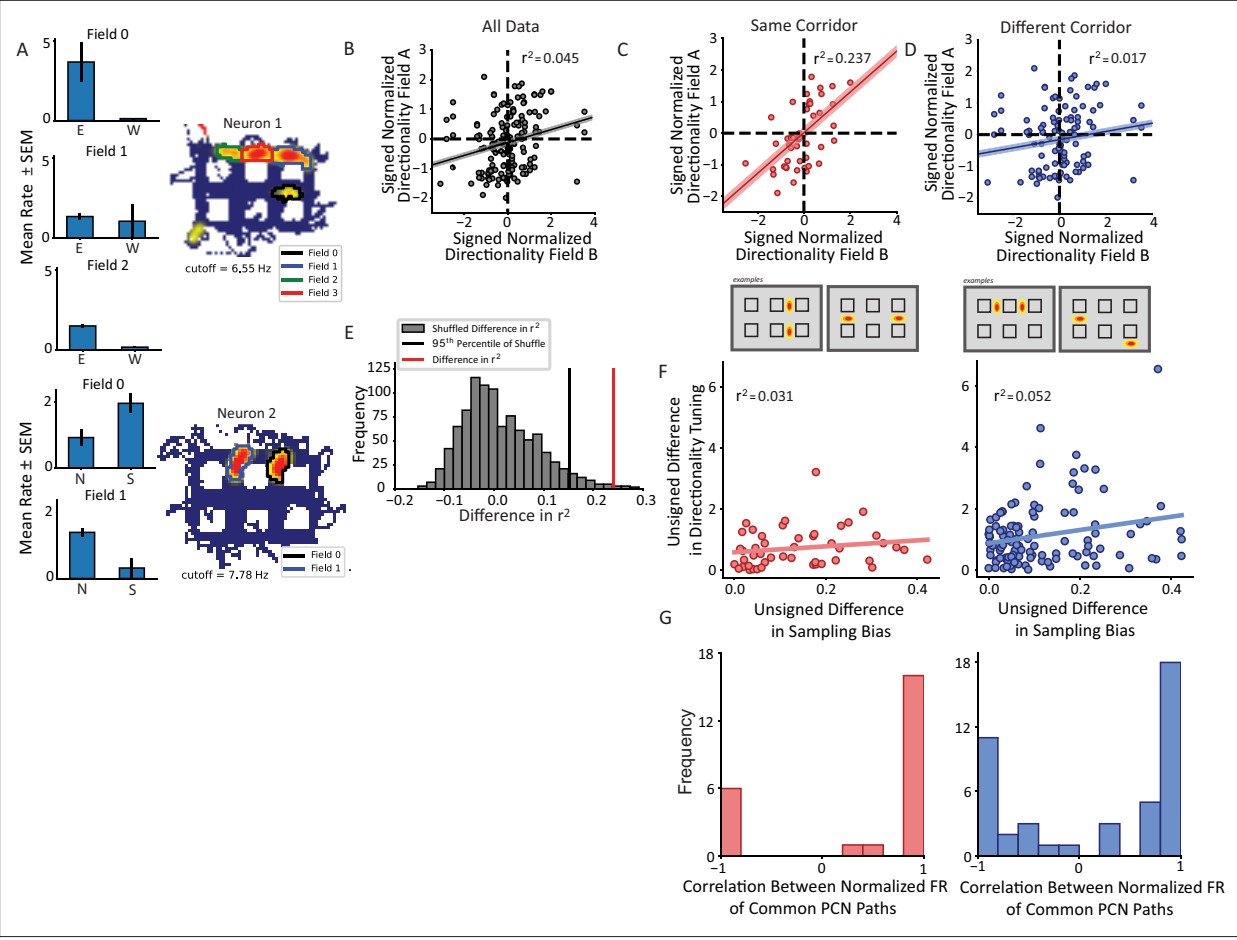

**Figure 5.** Nonconservation of repeating field directionality. (**A**) Examples of repeating neurons with directional tuning that was not conserved across fields. Top, neuron with four fields in horizontal alleys, three of which had sufficient sampling for analysis. Fields 0 and 2 preferred east over west while field 1 was untuned. Bottom, two fields primarily in vertical alleys that display opposite tuning patterns such that field 0 fires more strongly for south and field 1 for north. (**B**) The pairwise correlation in directionality tuning between all pairs of aligned repeating fields within the same cell. The directionality index for each field was normalized by the field's maximum. The correlation between points explains little of their variance ($r^2$=0.045, p=0.007). Pairs of fields were overall not much more likely to share (i.e. Quadrants I and III) as to not share (i.e. Quadrants II and IV) directional tuning. (**C, D**) Field pairs were broken down based on whether the fields were on the same corridor or different corridors. Two examples of each type of relationship are shown (bottom schematics). North and east were taken as positive, by convention. Field pairs on the same kind of segment had a higher correlation ($r^2$=0.237, p=0.00012) than those that were on different segments ($r^2$=0.017, p=0.125). The difference between the correlations was significant using a Fischer's r-to-Z transformation (independent sample, one-sided test, z=2.49, p=0.0063). (**E**) The difference in panels C–D was corroborated using a shuffling procedure. The labels for field pairs were shuffled while maintaining sample size. The difference $r^2_{same\_shuffle} - r^2_{different\_shuffle}$ was computed. The 95th percentile of this distribution was 0.155 which was less than the observed difference of 0.219. (**F**) The pairwise comparison of sampling bias and directional tuning between fields on same or different corridors. The sampling bias explained little in the pattern of directionality differences between fields in either corridor group. (**G**) Correlation between average response to each trajectory shared by fields comprising pairs on same (red) or different (blue) corridors showed bimodal distributions. A Hartigan's test showed that the distributions were not unimodal.

The online version of this article includes the following figure supplement(s) for figure 5:

**Figure supplement 1.** Corridor analysis controls.

**Figure supplement 2.** Lack of differences between male and female rats.

hypothesized that repetition reflects a common tuning among the fields to some path-related parameter (*Frank et al., 2000*). Alternatively, it is possible that at least some repeating neurons receive spatially modulated directional inputs, and thus repeating fields from the same neuron may not always share a common directional tuning. Such a finding would suggest that the positional and nonpositional aspects of a place field's firing are dissociable.

To compare these alternative possibilities, the pattern of directional firing across repeating fields was tested. Fields within each repeating neuron were separated by orientation to equate the directions

being compared. Directional tuning was not always conserved across repeating fields. For instance, neuron 1 of *Figure 5A* had four fields in horizontal alleys, three of which had sufficient sampling for analysis. Fields 0 and 2 fired more strongly for east over west while field 1 was untuned (two-way ANOVA interaction field x direction F(2) = 5.32, p=0.006). Neuron 2 had two fields in vertical alleys with opposite tuning. Field 0 fired more strongly for south while field 1 did so for north (two-way ANOVA interaction field x direction F(1) = 9.79, p=0.0025). To understand the distribution of directionality across repeating fields, a pairwise comparison was made between the directionality index of a given field and all other fields with the same orientation from the same cell and these comparisons were pooled across the population. The directionality index was calculated as the difference in the average firing rates between travel in either direction, normalized by both the mean firing of the field and location of each pass in the field. This distribution was centered at zero with density extending into all four quadrants, with a slight bias for fields to share the same tuning (113 pairs sharing, 81 not sharing directional tuning; chi-square goodness of fit test $\chi^2(1)$=5.28, p=0.022; *Figure 5B*). The polarity explained very little of the variance in the data (ordinary least squares [OLS] $r^2$=0.045, p=0.007). Thus, although there was a tendency for repeating fields to share the same directional bias, the size of the effect was small, demonstrating that repeating fields overall are weakly correlated in their directional tuning.

Given the role of borders and local geometry on firing within the hippocampus and its inputs (*O'Keefe and Burgess, 1996*; *Skaggs and McNaughton, 1998*; *Frank et al., 2000*; *Savelli et al., 2008*; *Savelli et al., 2017*; *Liu et al., 2021*), it was hypothesized that the geometry of the apparatus may influence the pattern of directional firing within repeating neurons. Field pairs were divided into whether the fields comprising the pair were on the same corridor or different corridors of the maze. Corridors were defined as the linearly concatenated alleys running the length or width of the track (*Figure 5C and D*, bottom schematics). Field pairs from the same corridor had a significant positive correlation in their directional tuning (OLS $r^2$=0.237, p=0.00012, *Figure 5C*). However, pairs on different corridors were not correlated with one another (OLS $r^2$=0.017, p=0.125, *Figure 5D*). These correlations were significantly different from one another (Z=2.49, p=0.0063, one-sided independent sample Fisher's Z-transformation test). These data suggest that the weak positive correlation in the total population of pairs is explained by a stronger correlation among the same corridor field pairs alone. This effect was corroborated by a shuffling approach. Labels of each field pair were shuffled while preserving the sample size of the two groups (same corridor, different corridor). The difference in $r^2$ was computed on each iteration between the two groups and the distribution of such values was compared to the real difference in $r^2$ values. The real difference of 0.219 was greater than the 95th percentile of the null distribution of differences which was equal to 0.155 (*Figure 5E*).

The effect of corridor on repeating field directionality is unlikely to be explained by behavioral sampling. *Figure 5F* quantifies the correlation in directional sampling between field pairs on the same corridor versus different corridors. For each field, the directional sampling bias was computed as the ratio of the number of passes in the direction sampled more frequently to the total number of passes. The unsigned difference in the sampling biases of the fields was compared to the unsigned difference in the field's respective directionality tuning. Very little variance in the pairwise directionality comparison was attributable to the differences in sampling bias (same corridor $r^2$=0.031, p=0.23; different corridor $r^2$=0.052, p=0.016), suggesting that the observed results were not driven primarily by imbalanced sampling of preferred and anti-preferred directions of directionally tuned fields.

The distances between fields forming a pair on the same corridor (17.55±7.56 bins) were similar to fields on different corridors (20.1±8.27 bins), although the small difference was statistically significant (Mann-Whitney U=2097, p=0.031; *Figure 5—figure supplement 1A*). This result suggests that a widely tuned spatial input is not the primary driver of the dissociation in directional correlations. The path taken to reach each field of a pair on the same corridor was shorter (4.17±2.54 alleys) than between fields on a different corridor (8.01±3.47 alleys) (Mann-Whitney U=607, p=8.04 × 10^-11; *Figure 5—figure supplement 1B*), raising the possibility that the trajectories between the fields systematically differed and this difference affected the shared directional tuning.

Last, we investigated whether a more specific analysis of trajectories would show a tendency for repeating cells to show trajectory coding. Trajectories were defined as sequences of directions (previous, current, and next) through the fields (e.g. east-north-east, south-west-north, etc.) and each trajectory through a field was assigned to such a trajectory. For each pair of commonly oriented fields,

the correlation coefficient was calculated between the average responses to each type of trajectory sampled through both fields. This analysis revealed a non-unimodal distribution (Hartigan's dip test statistic = 0.122, p<2.2 × 10⁻¹⁶, corridor conditions pooled together; *Figure 5G*), with over half of all field pairs having a positive correlation (56% pairs correlation>0.5). Therefore, although a majority of field pairs tended to share a pattern of trajectory responses, a substantial fraction did not. Among the latter pairs, the responses were mostly anticorrelated (i.e. the correlations were close to –1). These data suggest that field pairs are affected by the trajectory through the field, since they tended to respond to the same trajectory – either by sharing their firing or having opposite responses. By contrast, very few field pairs were uncorrelated in their response to the same trajectory.

## Time dynamics of CA1 firing rates

CA1 activity patterns change across disparate timescales (*Pastalkova et al., 2008*; *MacDonald et al., 2011*; *Manns et al., 2007*) in ways that may support the encoding of temporal context (*Ziv et al., 2013*; *Cai et al., 2016*; *Mankin et al., 2012*). Prominent temporal non-stationarities were observed in the present study, taking on a number of different qualitative patterns (*Figure 6A*). *Figure 6A* (top) shows an example of a single-field cell that decreased its firing rate over the course of the session. Other cells (not shown) increased their firing rates over the session. Intriguingly, repeating fields of the same cell also showed temporal dynamics that were decoupled in time. *Figure 6A* (middle) shows a cell with two repeating fields that showed antiphase temporal dynamics for the first 40 min (i.e. the firing rates of each field waxed and waned out of phase with each other). *Figure 6A* (bottom) shows a final example of a repeating cell with four fields (three horizontal and one vertical) that each showed a rapid increase in firing rate followed by a slow decline, but the temporal onset varied across the fields.

Across all cells, a GLM approach identified a significant number of fields that exhibited significant time dynamics (83/296 oriented fields; binomial test, p=2.07 × 10⁻³⁸; average increase in fit 0.26±0.38; *Figure 6B*). The GLM result was corroborated with a randomization test which shuffled the time labels of field visits and repeated the model comparison process; this procedure resulted in 5% of fields with a fictive pattern of time dynamics, well below the experimental value of 28% (*Figure 6—figure supplement 2*). Repeating and non-repeating fields were equally likely to exhibit temporal dynamics, demonstrating a similar lack of difference seen in directional tuning (66/171 repeating fields, 39/125 non-repeating fields, Pearson's chi-square $\chi^2(1)$=0.53, p=0.42). Fields from the same repeating neuron were also analyzed to see if their time dynamics were correlated. The Pearson's correlation between the firing rate time series was computed between each pair of fields of each repeating neuron (*Figure 6C*). This distribution was centered at zero, suggesting fields were not more likely to be correlated or anticorrelated with one another (mean correlation = 0.049 ± 0.26).

These time dynamics resulted in gradual changes in the population activity over time. Data from each recording day were split into six windows and the population vector correlations between windows were calculated. More distant windows were less correlated (average slope per dataset –0.06±0.022, two-sided t-test compared to 0, t(9) = 7.88, p=2.49 × 10⁻⁵; *Figure 6D*), consistent with prior work (*Manns et al., 2007*; *Mankin et al., 2012*). This trend was maintained within both repeating and non-repeating groups separately (*Figure 6—figure supplement 1*). To explicitly test the change in positional information over time, a sliding window was moved across the session and the data in each window was used to train a linear regression classifier. The data in the training window was tested on all other windows (*Figure 6E*). This analysis revealed that the decoding performance was high near the training window and decayed outside of it, yielding a positive slope (mean Spearman correlation = 0.44 ± 0.49, one-sided t-test (greater), t=7.59, p=5.11 × 10⁻¹¹) to the performance in the window before the training window and a negative slope (mean Spearman correlation = –0.45 ± 0.43, one-sided t-test (less), t=10.0, p=1.53 × 10⁻¹⁶) to the performance after the training window (*Figure 6F*). Together, these data are consistent with previous work showing that time dynamics within CA1 are graded at the population level (*Manns et al., 2007*; *Mankin et al., 2012*), which is theoretically capable of providing a temporal context signal (*Ziv et al., 2013*; *Cai et al., 2016*; *Tsao et al., 2018*).

## Discussion

Analyzing the travel direction through place fields, both repeating and non-repeating, provided insight into three topics of CA1 physiology and its potential relationship to episodic memory formation

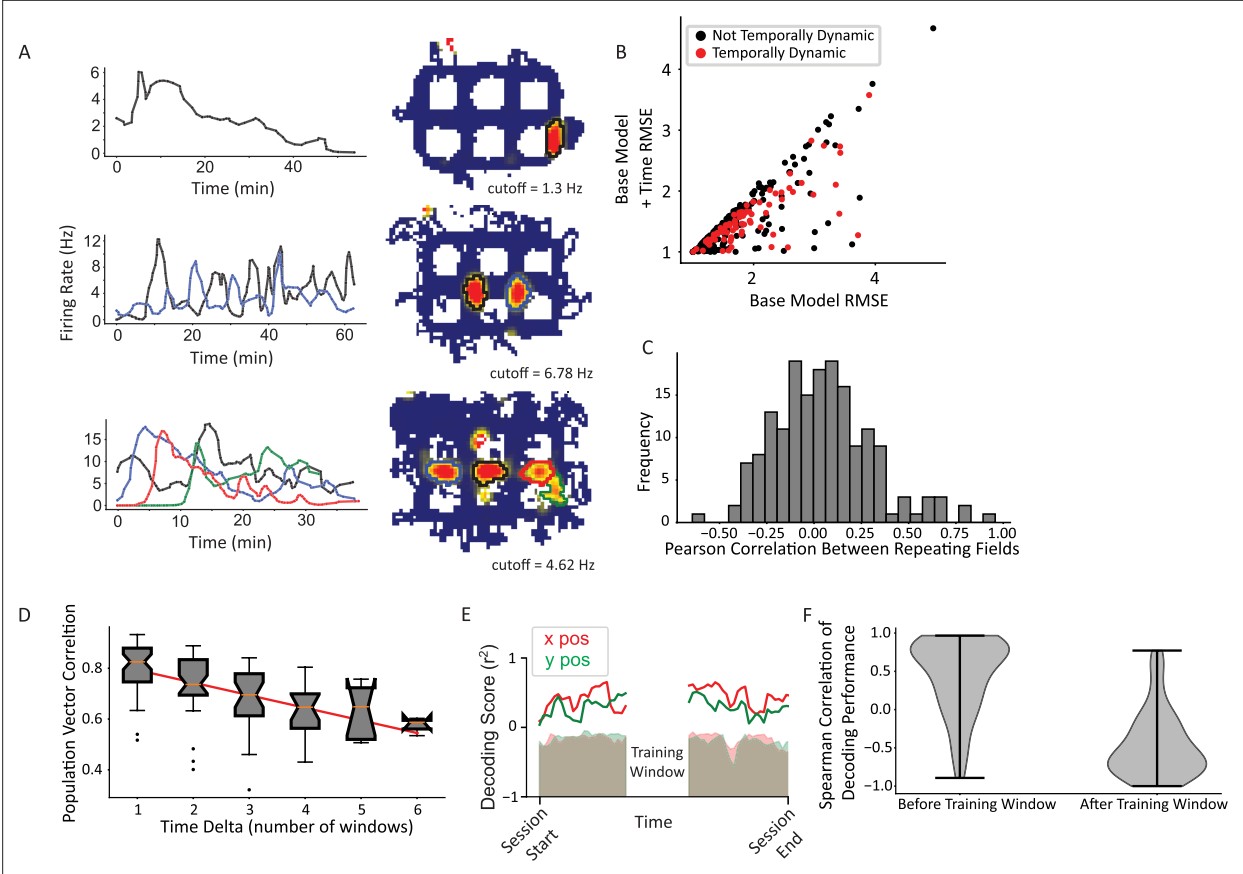

**Figure 6.** Temporal dynamics. (**A**) Examples of neurons with temporally dynamic place fields. Top, neuron with single field in a vertical alley. The field rate increases to a peak that is maintained for roughly 10 min, before slowly decaying toward near silence over the rest of the session. Middle, repeating neuron with two fields in vertical alleys. The fields appear to have complex periodic dynamics over time. Further, they appear to be in an approximate antiphase relationship in approximately the first half of the session, such that when the west field (black) is more active, the east field (blue) is at a local trough and vice versa. Bottom, repeating cell with four fields, three of which are in horizontal alleys. Each field peaks with a different delay and then gradually decays over time. The delays are such that the times during which each field is active tile the whole session and the cell maintains its overall activity over time. (**B**) A generalized linear model (GLM) likelihood ratio test to determine whether fields are temporally dynamic. The base model included current direction, for an analogous reason as to why time was included in the base model when testing for current direction. Time was represented as a natural spline with three degrees of freedom. A significant number of fields changed their firing rates over time (83/296 field pieces, $p=2.07 \times 10^{-38}$, alpha = 0.0083 for three spline knots multiplied by (up to) two orientations per field). (**C**) Pairwise comparisons of the firing rate time series of pairs of repeating fields were computed using Pearson's correlation. The firing rate time series were interpolated to equate their sample sizes using a piecewise cubic Hermite interpolating polynomial (PCHIP) interpolation. The distribution was centered near zero with a wide distribution (mean = 0.049 ± 0.26). (**D**) Pairwise correlations between activity patterns in different time windows. Time windows closer together are more correlated than those further away. There was no significant difference in the slope of the decorrelation between repeating and non-repeating neurons (*Figure 6— figure supplement 1*). (**E**) Example time window from single dataset showing portion of data held out for training (white space, center) and the decoding performance on the rest of the data divided into temporal windows. Shaded region is shuffle. Red corresponds to x position data, green to y position data. (**F**) Distribution of Spearman correlation of decoding performance before or after the training window shows the gradual decrease in decoder performance with increasing distance from the training window.

The online version of this article includes the following figure supplement(s) for figure 6:

**Figure supplement 1.** Temporal drift in repeating versus non-repeating neurons.

**Figure supplement 2.** Shuffling test of temporal dynamics.

according to *O'Keefe and Nadel, 1978*, cognitive map theory. First, the presence of directional firing on a maze that had properties of both open-field foraging and directed navigation tasks suggested that the low degrees of freedom for path choices resulted in the system distinguishing between the paths, possibly because they differed from one another in limited ways constrained by the geometry of the apparatus rather than continuously, as they would on an open platform. Second, comparing directional tuning across repeating fields revealed a distinction between positional inputs (i.e. inputs

that tell a place cell *where* to fire) and nonpositional inputs (i.e. inputs, including spatial inputs, that tell a place cell *how strongly* to fire in its place field). Last, in spite of the spatial ambiguity introduced by multi-fieldedness and temporal dynamics, the map maintained a stable spatial signal but exhibited gradual temporal dynamics, with repeating fields showing temporally dissociable changes similar to their dissociation in directionality.

## Directionality, trajectory coding, and place field repetition

Place fields are more directional when animals perform directed, stereotyped trajectories on unidimensional tracks (*McNaughton et al., 1983*; *Markus et al., 1995*; *Liberti et al., 2022*) compared to 2D foraging in an open field (*Muller et al., 1994*; *Markus et al., 1995*; but see *Rubin et al., 2014*; *Acharya et al., 2016*). The city-block maze of the present study had aspects of both types of behavior, as the rats could move in 2D, but their trajectories were constrained to a rectilinear grid. Nonetheless, many place cells showed strong directionality and, furthermore, showed large numbers of repeating fields. This combination of place field repetition and directionality has been interpreted as evidence for trajectory encoding, or 'path equivalence', in which spatially selective neurons in the hippocampus or entorhinal cortex fire in similar locations along a trajectory that is repeated at different locations in an environment. The data presented here suggest that trajectory encoding by repeating fields is likely to be more complex than previously characterized (*Frank et al., 2000*; *Singer et al., 2010*). In the previous work, trajectory coding was explored on apparatuses with a constrained set of choices (*Frank et al., 2000*; *Singer et al., 2010*; *Grieves et al., 2016b*; *Ferbinteanu and Shapiro, 2003*; *Smith and Mizumori, 2006*; *Wood et al., 2000*). These studies found neurons tuned to different types of turns and they suggested that more complex trajectories might be encoded on apparatuses where such trajectories could be sampled. The present work utilized such an apparatus and provided qualified support for this suggestion. Due to the unstructured nature of the task, the data lacked the power to test whether any particular complex trajectory (defined as a sequence of travel directions into, through, and out of an alley) was encoded by individual neurons. Thus, current direction alone was used as a simple proxy for the potentially complex path through the field. If place field repetition reflected the hippocampus encoding equivalent paths in an environment, then the different fields of a repeating cell should show the same current direction preference. Although previous work has demonstrated a role for direction in gating the presence of repetition (*Grieves et al., 2016a*; *Harland et al., 2017*; *Cowen and Nitz, 2014*; *Derdikman et al., 2009*), it remained unclear if a common representation of direction was required for repeating fields. In the present work, many repeating fields were uncorrelated in their responses to direction (at least when the repeating fields were on different corridors), showing that directional preference per se was not the driving force underlying repetition under the conditions of the present study.

Although the lack of shared directional preference in many neurons suggested that these neurons were not encoding trajectories in a simple manner, we nonetheless observed that the repeating fields of many cells showed strong correlations in their firing rates across the different types of complex trajectories through the fields (i.e. previous, current, and next alleys) (*Figure 5G*). This result suggested that a subset of repeating fields may indeed encode trajectory. However, many repeating fields in different corridors were anticorrelated in their trajectory responses. Thus, place field repetition may result from multiple mechanisms that give rise to similar phenomena: a subset of repeating fields may encode trajectory while another subset encodes a different dimension, such as local border geometry.

The potential contribution of local border geometry to place field repetition is supported by multiple studies and was specifically proposed as a mechanism in one modeling study (*Grieves et al., 2017*). In the present work, as well as others recording from multiple compartments (*Skaggs and McNaughton, 1998*; *Spiers et al., 2015*; *Grieves et al., 2016b*), the local border geometry was shared across fields of multi-fielded neurons (*Figure 2D*). Cells in the present work were disproportionately likely to have fields with borders on the same sides of the field, i.e., to be aligned along an axis of the maze (vertical or horizontal) (this is similar to axis-tuned cells reported by *Olson et al., 2017*). In previous work, fields occurred in the same relative position in multiple adjacent boxes (*Skaggs and McNaughton, 1998*; *Spiers et al., 2015*; *Grieves et al., 2016a*). This hypothesis is consistent with the inputs that CA1 receives. Boundary vector cells (BVCs) (*Lever et al., 2009*; *Stewart et al., 2014*) and border cells (*Barry et al., 2006*; *Savelli et al., 2008*; *Solstad et al., 2008*) exist within the circuitry of the hippocampal formation and it is known that CA1 neurons are modulated by changes to the local

border configuration (*O'Keefe and Burgess, 1996*). Thus, it is plausible that repetition may be caused by stronger border-related input to these neurons. The paths available at a location are partially determined by the configuration of borders, therefore, the hypotheses discussed above are related. Future work would require the dissociation of the pattern of borders and the types of paths available to the animal at each location to disentangle the effect of these variables.

## Leveraging repetition to understand positional versus nonpositional inputs

Place field repetition offers a unique opportunity to explore how inputs are functionally organized in CA1. According to the cognitive map theory (*O'Keefe and Nadel, 1978*), the hippocampus leverages the spatial framework of the place map to bind together different aspects of an experience into a coherent representation in support of episodic memory storage and retrieval. The present data provide evidence that this process is assisted by individual field 'addressability', wherein different nonpositional inputs (i.e. experiential inputs) affect a neuron's response differently at different fields. Our population decoding analyses demonstrated that both the positional signal and the nonpositional signal could be retrieved from such a multiplexed coding scheme.

The rate remapping hypothesis stipulates a mechanism utilizing the rate remapping phenomenon to encode nonpositional inputs – such as changes to the sensory features of an environment or differences in how the animal navigates through the environment – through changes to the pattern of firing rates across spatially stable place fields (*Leutgeb et al., 2005*; *Shapiro et al., 1997*). This rate remapping allows for flexible encoding of the episodic content of experience while maintaining a stable map. Such episodic content may include nonspatial variables, such as alterations in objects or the wall colors of an environment, or spatial variables that are related to navigation but do not define specific positions. For example, direction and speed can alter the firing rates of place cells (*McNaughton et al., 1983*; *Markus et al., 1995*; *Czurkó et al., 1999*; *Maurer et al., 2012*; *Huxter et al., 2003*), and fields of 'splitter cells' can alter their firing rates depending on the upcoming (or previous) spatial choice, while the field locations remain mostly stable (*Wood et al., 1999*; *Frank et al., 2000*; *Ferbinteanu and Shapiro, 2003*). These results suggest that inputs that are spatial but do not explicitly encode position may be encoded similarly as purely nonspatial inputs such as odors or sounds. Thus, we have used the term nonpositional (as opposed to nonspatial) to denote how various cues can alter the firing rates of place cells while maintaining the positional specificity of the place field intact.

Repeating place fields provided a within-cell control to demonstrate that positional inputs (e.g. functional influences that tell a place field where to fire) and nonpositional inputs (i.e. functional influences that tell the neuron how strongly to fire in its field) are organized differently in CA1. The simplest model would suggest that both sets of inputs are fixed for each place cell. Thus, the positional inputs would cause a cell to fire at similar locations in the environment (e.g. locations that share common elements of a trajectory or that share common geometric borders) and share the same nonpositional coding (e.g. similar direction preferences). However, such inflexibility would not be ideal for the hypothesized role of firing rate as encoding the episodic inputs onto a place field, which would require flexibility in encoding unpredictable experiential inputs at different locations. The present data utilize repeating fields to suggest that different repeating fields can show different directional preferences. In other words, fields do not inherit their nonpositional tuning as a global property of the cell.

In different environments, the firing locations of place fields change when the system undergoes global remapping. This well-studied property suggests that the positional inputs can also be dynamically gated and flexible between environments. On the other hand, it is possible that the positional inputs are also rigid between environments, even as the place fields express different firing fields. For example, the BVC model (*O'Keefe and Burgess, 1996*) postulates that each place cell receives input from a fixed set of BVCs that determine whether and where a cell will fire in an environment. In this model, a place field will be active if there is a location in the environment that drives each of its BVCs. Thus, it is the environmental geometry that determines whether a cell will show global remapping between environments, not flexible changes to its positional inputs (*Hartley et al., 2000*). Regardless of whether the positional inputs are rigid or flexible across environments, the present data suggest that the nonpositional inputs are always flexible, as expected if they encode the episodic content of an experience on a stable spatial framework.

The corridor analysis (*Figure 5C and D*) revealed that under certain conditions the nonpositional input tended to be shared across repeating fields, suggesting that experience may shape directional tuning. The difference in pairwise directional tuning between fields on the same and different corridors further emphasizes that fields were separately addressable in how they were tuned for direction. The difference in corridor was also correlated with behavioral biases in how animals sampled fields. Fields on the same corridor were usually sampled by shorter paths (*Figure 5—figure supplement 1B*) that did not necessarily involve a change in direction of travel. However, traveling between fields on different corridors necessarily involved a change in direction and these paths were usually longer as well (*Figure 5—figure supplement 1B*). One explanation for these data is that fields sampled along contiguous trajectories, without interruptions from direction change or reward delivery, are more likely to share their directionality. Directionality is affected by an animal's experience through the field (*Navratilova et al., 2012*), so it is possible the difference in experience between sampling fields on the same versus different corridors affects the directional tuning properties between them. Another possibility is a shift in reference frames (*Gothard et al., 1996*; *Jackson and Redish, 2007*; *Fenton et al., 2010*). In this scenario, place cells respond differently depending on which of multiple reference frames the ensemble (or subset thereof) is locked to. The transitions between alleys along different corridors may induce or somehow be associated with such a reference frame shift. Sequence learning mechanisms, such as theta phase precession or theta-gamma coding (*Gothard et al., 1996*; *Lisman, 2005*; *Terada et al., 2017*) may also play a role. It is known that theta look-ahead sequences, in which non-local place field activity sweeps ahead of the animal, segment behavior by preferentially occurring at landmarks (*Wikenheiser and Redish, 2015*; *Gupta et al., 2012*). In the present study, turns at the ends of corridors, along with reward deliveries, may be salient task boundaries at which point theta sequences are terminated, although we lacked a sufficient number of simultaneously recorded repeating neurons to test this directly (cf *Wikenheiser and Redish, 2015*). Fields active within the same theta sequence (typically same corridor fields) may be functionally coupled, while fields active on opposite sides of a theta sequence termination (different corridor fields) may be uncoupled and their tuning uncorrelated.

## Temporal dynamics and map stability

The present data demonstrate a constant, gradual drift in the population activity pattern as the animal forages without a task. This affects the stability of the map over time; although the map maintains spatial informativeness at each point in time, the activity of the neurons comprising the map exhibits strong temporal non-stationarities. The time dynamics within individual repeating fields were heterogeneous and uncorrelated. These time dynamics do not necessarily correspond to the encoding of time *per se*, and further experiments would be needed to make that claim. For example, unknown variables that change over time may be altering the firing of these cells. However, the relevance of time dynamics for the present study is that the fluctuations over time were not always synchronized across repeating fields. This provides evidence for another nonpositional signal, time, or some temporally varying signal, that is not conserved across fields. However, it is also important to note that for most repeating cells the positional input was stable in that it provided a consistent instruction for where to fire as the field drifted; fields tended to drift between analogous locations. Previous studies have shown temporal drift hypothesized to serve as a temporal context signal to distinguish similar, repeated events (*Manns et al., 2007*; *Ziv et al., 2013*; *Mankin et al., 2012*). However, these studies did not show a constantly drifting signal in the absence of an explicit task, which would be predicted if these dynamics served as a temporal context signal. One study did measure representational correlation in the absence of a task (*Kentros et al., 2004*), but analyzed similarity across sessions separated by at least 30 min. Here, we observed slow drift during continuous recordings on the scale of minutes to tens of minutes. In one study examining temporal drift during a foraging task, only two short-term timepoints were used which made it difficult to determine whether there was a continuous signal (*Mankin et al., 2012*). In another study (*Manns et al., 2007*), drift was analyzed over shorter periods, but during an odor discrimination task which may recruit additional circuits than what would be used during free navigation. Therefore, the present data show that CA1 activity patterns are constantly drifting even in the absence of a task (as demonstrated previously in LEC; *Tsao et al., 2018*) but in a way that preserves the spatial informativeness of the map.

The uncorrelated time dynamics within repeating fields provides further evidence that place cells do not receive a global gain signal that affects all fields equally. Fields changed at different times (*Figure 6A*) and at the population level this resulted in decorrelated repeating field time series (*Figure 6B*). It is important to note that because a Pearson's correlation was used, it is possible the fields are related in time with a phase shift, e.g., the middle row of *Figure 6A* shows repeating fields that appear to be in antiphase with each other for approximately 40 min while the bottom row of *Figure 6A* shows repeating fields that suddenly increase their firing at a 5–6 min delay with a subsequent slow decay of each field. However, because the heterogeneity of temporal responses among fields was so large, it was difficult to develop a quantitative measure that captured this diversity in a way that would allow statistical evaluation. Thus, although it is clear that the place fields of repeating cells do not change their firing rates in synchrony, as if the cell had a global excitability change that made all its fields wax and wane together (e.g. from some neuromodulatory input or network effect on the scale of seconds to minutes), it nonetheless remains an open question as to whether the subfields of repeating cells engage in certain types of competitive interactions or other network dynamics that couple changes in their firing rates in more complex ways.

## Open-field foraging versus directed navigation

The behavioral sampling of an environment may affect how the CA1 place map represents it. CA1 place cells distinguish between different behaviors as the animal traverses a field if those behaviors are relevant for goal-directed navigation. When rats were trained to run down the stem of a T-maze before making a choice, some neurons on the center stem common to both choices distinguished between the behaviors (*Wood et al., 2000*). These cells, termed 'splitter cells', have been recorded under a variety of circumstances (*Ferbinteanu and Shapiro, 2003*; *Smith and Mizumori, 2006*; *Bower et al., 2005*; *Griffin et al., 2007*; *Grieves et al., 2016b*, though see *Lenck-Santini et al., 2001*). Common to these directed navigation experiments were stereotyped behaviors the animal learned, typically trajectories from a starting position to a goal. Place fields can also be directional, with their firing rates changing as a function of the direction of travel through the field (*McNaughton et al., 1983*; *Wiener et al., 1989*; *Markus et al., 1995*). Although it is possible directionality is caused by other correlates of direction change, e.g., changes in spatial view (*McNaughton et al., 1991*), another interpretation is that directionality and splitting both reflect common, or at least similar, mechanisms that cause place fields to fire differentially depending on the path through the field. For example, place fields became directional when animals were trained to run on an open platform in a stereotyped manner (*Markus et al., 1995*), conditions that are similar to those that elicit splitting and result in a qualitatively similar phenomenon. The conditions under which splitting/directionality are observed are not well understood and the presence of directionality shown here suggests that the pattern of behavioral sampling per se might be sufficient to cause splitting/directionality. The present design is a hybrid between the open foraging-like conditions, in which little directionality/splitting are seen, and the directed navigation tasks, in which splitting is observed. It is important to note that the rats did not explore the maze randomly (*Figure 1—figure supplement 2*) and the design has elements of a directed search task as well given that rats had to shuttle between known, fixed sites to find reward. In the present design, there were no specific paths for the animal to learn; it could freely navigate throughout the city-block maze. However, its behavior was constrained by the linear nature of each alley and the right-angle choice points. This resulted in a low choice degree of freedom on the maze. The fact that a substantial number of place fields distinguished between the directions of travel through their firing rates suggests that even without learned trajectories to perform, CA1 represented attributes of the path through the field. It is possible that the rat performed internally generated trajectories, and these were represented by CA1. Alternatively, the low choice degrees of freedom itself could have promoted the representation of trajectory-related variables, possibly because of the discrete nature of travel through the maze. Regardless, these data demonstrate that explicit learning itself is not necessary for the hippocampus to represent how the animal moves through a maze.

## Materials and methods

### Animals and pretraining

Five Long-Evans rats (3 males, 2 females, 3–6 months of age, 325–350 g males, 215–250 g females) were used in the study. Data from male and female rats were consistent with one another and so data from all rats were pooled together (*Figure 5—figure supplement 2*). Animals were maintained at 80% of their ad libitum weight to provide motivation during the task. Pretraining began once the animal's weight was near the target weight and its behavior was calm in response to handling. Animals were trained once a day for approximately 1 hr to run clockwise laps around the perimeter of the track for a separate experiment, receiving liquid reward (~50 μL 1:1 water:Ensure, Abbott Laboratories, Abbott Park, IL, USA). Two rewards were given pseudorandomly per lap. Over the course of pretraining the number of laps and time spent in a start box were increased. When the animal reached a performance criterion of 30–60 laps of consistent, motivated behavior, the animal was deemed ready for hyperdrive implantation.

### Hyperdrive construction and implantation

Custom hyperdrives were implanted over right CA1 (2.5 mm posterior and 2.3 mm lateral to bregma, *Figure 1—figure supplement 1*) in each animal, allowing simultaneous recording of hippocampal ensembles from independently adjustable tetrodes (*Wilson and McNaughton, 1993*). The hyperdrive design consisted of a custom, 3D-printed body with 18 independent microdrives (16 tetrodes plus 2 reference electrodes). Each tetrode was composed of four 0.0017" diameter nichrome wires coated with VG and HML bonding agents (California Fine Wire Co, CA, USA), twisted together, and heated with a heat gun to bond the insulation. Each tetrode was electroplated to a target impedance of 130 kOhm from a typical starting impedance of 2–3 MOhm using a NanoZ plating system (NanoZ System, Multi Channel Systems, Reutlingen, Germany). The hyperdrive was gas sterilized with ethylene oxide for 12 hr before implantation.

Anesthesia was induced with 4% gaseous isoflurane (1 L $O_2$/min) followed by an injection of ketamine (80 mg/kg) and xylazine (10 mg/kg). Surgical anesthesia was thereafter maintained with isoflurane inhalation (to effect). Marcaine (0.15 cc) or ropivacaine (0.15 cc) were used as local analgesics applied directly to the skull. Meloxicam (2 mg/kg) was given as an analgesic via IP injection post-op and orally 24 hr after surgery. All animal procedures were approved by the Institutional Animal Care and Use Committee of Johns Hopkins University and complied with National Institutes of Health guidelines.

### Electrophysiological recordings

Tetrodes were advanced ~1.3 mm at the time of surgery and advanced further to the pyramidal layer of CA1 over the course of approximately 2 weeks using stereotypical changes in the EEG (specifically the theta and sharp-wave signals) as guides. All recordings were performed using a 64-channel Neuralynx Digital SX recording system and Cheetah v5 or v6 software (Neuralynx, Bozeman, MT, USA). Electrophysiological signals were filtered using a finite impulse response (FIR) filter from 600 to 6000 Hz for tetrodes and 1–300 Hz for the EEG. The low-cut taps for the FIR filter were 32 and 0 and the high-cut taps were 64 and 128, for the tetrodes and EEG, respectively. Thresholds for spike detection were set to a minimum of 50 μV. Spiking data were sampled at 32 kHz with each spike comprising a 1 ms sample (7 samples before the waveform peak, 24 after). EEG data were sampled continuously at 32 kHz.

### Spike sorting

Unit isolation was performed using custom software (WinClust, JJ Knierim). Spike waveform parameters detected from each tetrode wire were visualized against one another on a 2D plot. Given that the signals from each neuron impinge differently on each electrode wire, the spikes from each neuron appeared as different clusters on the plot. These clusters were manually circumscribed on multiple parameter projections and the intersection of these bounds formed the inclusion criteria for each unit. Each cluster was then subjectively ranked on a five-point scale according to how well it was isolated from both the background noise and other, neighboring clusters. Only clusters with an isolation score of three or better were analyzed further. For each rat, two datasets with the best behavior and highest number of well-isolated clusters were selected for quantitative analysis.

## Video tracking

Tracking of the animal's position was achieved using an LED mount attached to the hyperdrive adapter board. An overhead camera recorded data at 30 fps. Color and luminance thresholds were set each day before the experimental session. Positional information was calculated by taking the center of mass of a luminance-thresholded camera frame, while direction was found using a vector defined by the front and back LEDs (which were colored differently for identification).

## Behavioral apparatus

A novel city-block maze was constructed for these recordings (*Figure 1A*). The track consisted of a 0.9×0.6 m$^2$ walled platform with six blocks (18.4×18.4×15.2 cm$^3$) arranged in a 2×3 grid. The gaps between the blocks and either the other blocks or the perimeter wall created 17 alleys (14×8.25 cm$^2$). An IR emitter/detector (emitter: Honeywell, Charlotte, NC, USA; detector: Lite-On, Taipei City, Taiwan) pair was affixed to both entrances to each alleyway. A lickport connected to a peristaltic pump (Adafruit, New York, NY, USA) was mounted to the wall of each alley with a visible light LED mounted above and a buzzer below the surface of each alley. The hardware and software for each alley were controlled by a separate Arduino Uno (Arduino, Somerville, MA, USA), while the task was controlled by a custom Python program connected to the track via an Arduino Mega and communicating with each alley via the I2C protocol (*Moelands and Schutte, 1993*).

## Behavioral paradigm

Animals navigated freely within the city-block maze for roughly 1 hr each day, with 15–20 min of baseline recording in a small holding dish before and after recording. Animals had no prior experience on the interior of the maze before recordings started. Liquid reward (75 µL 1:1 Ensure:water) was dispensed from lickports affixed to the face of each of six blocks defining the interior structure of the maze. When infrared beams detected that the animal crossed the threshold of an alley primed for reward, a light flashed and a tone played while reward was dispensed. Two rewards were available on the apparatus at any one time, with a new reward site selected pseudorandomly once a primed reward site had been visited. The reward selection was performed such that a new reward site must be at least two alleys distant from the site it was replacing, to encourage sampling of the entire apparatus.

## Place field detection

2D ratemaps were created with a bin size of 10 pixels (2.1 cm) and a Gaussian smoothing kernel of 1.5 pixels. To identify place fields, local firing rate maxima were found with a minimum distance of 40 pixels between them and the borders around each were expanded until the firing rate decreased below 20% of the peak rate for each bin of the expanding border. These regions were then passed through two thresholding steps. First, the area of the field had to be ≥1% of the walkable track area. Second, the peak firing rate had to be ≥1 Hz. Regions around a peak that did not pass these thresholds were not included for analysis; this was done to filter small, weakly active regions. Each field was then iteratively split into smaller fields using a watershed algorithm using skimage.morphology. watershed (subsequently moved to skimage.segmentation.watershed; *van der Walt et al., 2014*). This splitting process identified all local maxima within the field. For each pair of in-field local maxima, the watershed algorithm expanded the regions around each peak until they collided, resulting in two potential place fields and the boundary between them. These smaller fields were considered valid if they passed the size and activity thresholds described above and if the firing rate of the boundary between them was not more than 75% of the firing rate of the larger peak. If the smaller fields did not pass these thresholds, they were kept as a single, larger field comprising them both. Each detected field of each neuron was assigned to three possible types of regions: a vertical alley, a horizontal alley, or an intersection. Assignment was determined based on overlap between the area of the field and the area of the track region. If the field overlapped at least 30% of the area of the track region (intuitively, it encroached into an alley at least 30% of the alley's length), then the field was defined as being located in that region. If the field overlapped multiple locations, each overlap location was tested to see if the section of field in that location contained at least 25% of the field's overall firing rate. This was computed by summing all the firing rate values in the bins of the ratemap in the region and dividing by the summed rate for the whole field. The field was labeled as being present in any and

all locations that passed this measure. A manual curation step was also performed (see 'Data filtering and preprocessing' below).

## Data filtering and preprocessing

The ratemaps described in the 'Place field detection' section were created with data that were filtered by running speed with a threshold of 1.5 cm/s. This threshold was chosen based on the structure of the environment, which forced the animal to make frequent turns and therefore had little opportunity to reach high speeds. Because the watershed algorithm may produce split fields that do not satisfy a firing rate criteria, a post-watershed firing rate threshold was applied to all fields. To be included in the dataset, the 95th percentile of the firing rate distribution of 2D ratemap bins must be ≥1 Hz. To exclude putative interneurons, any cell with a log mean rate >0.6 was excluded (*Figure 1—figure supplement 9*). The place field detection algorithm required manual curation. Field boundaries that did not correspond to a clear, on-track location of spatially tuned firing were excluded. In particular, periods of off-track exploratory head movements were removed at this step. Fields with boundaries that appeared poorly drawn, typically being overly large, were manually restricted. Importantly, this manual curation only removed poor quality fields, consolidated multiple detected 'fields' that were subjectively better explained by being peaks of the same field, or improved their boundaries; new fields were not created. See *Figure 1—figure supplement 8* for details.

Passes through the field were defined as consecutive camera frames within the boundary of the field. Samples that exited the field and re-entered within 2 s were grouped together as part of the same pass through that field. Further, the pass was required to visit at least two points inside the field and at least two of these inside points must have been far enough inside the field that the distance from the closest point of the field border to the visit sample was at least 5% of the distance along the shorter of the length or width of the field. The length was taken as the extension of the field along the y-axis and the width was the extension along the x-axis. Additionally, passes the animal made through each alley were filtered to exclude occasions when the animal turned around without fully traversing the alley. This occurred frequently when the animal appeared to check an alley for reward and leave the way it came if no reward was given. Passes on which a reward was delivered were also removed, in order to remove potential firing rate confounds due to reward. Major analyses were re-analyzed with all trials, rewarded and non-rewarded, to confirm that this did not affect the conclusions of those analyses (*Supplementary file 1*). Lastly, fields were excluded if there were fewer than two passes each in both directions through the alley. These preprocessing steps were not applied to the analysis of the number of track regions each field overlapped (see first paragraph of *Place field repetition* in Results) because this analysis pertained to the overall size and shape of the field and not the sampling through it.

Each pass was normalized by the mean activity at that location throughout the duration of the session. To do this, the spatial bins visited during the pass were identified and the occupancy-normalized firing rate in each spatial bin was computed using the data from a single pass. This single-pass ratemap was normalized by the session-averaged ratemap. The normalized pass firing rate was then defined as the mean of all spatial bins of the normalized single-pass ratemap. The rationale for this normalization was to control for passes which impinged on different parts of the spatial firing field. For instance, the animal may tend to sample the edge of the field when performing one type of trajectory (e.g. making a right turn from one alley to the next) and the center of the field when performing a different type of trajectory (e.g. moving straight ahead from one alley to the next). This would result in lower firing rates for the former behavior, even if the neuron did not in reality distinguish the two conditions.

As mentioned above, neurons could be classified as present at multiple alleys or intersections if their field overlapped a significant proportion (30%) of that region. For most of the analyses here, fields present at multiple locations were filtered by orientation (vertical or horizontal alleys). All analyses where such orientation filtering was performed were adjusted for this multiple comparison via a Bonferroni correction.

## Quantifying place field repetition

Place field repetition was defined in two ways, operationally and statistically. Operationally, place field repetition was defined as any neuron with two or more fields in a common type of location.

The common locations were vertical alleys, horizontal alleys, and intersections. Only alley data were analyzed here, except for the time dynamics data (*Figure 6*) which included alleys and intersections. This definition implies that any neuron with three or more fields in alleys was defined as repeating. Therefore, operationally, repetition was multi-fieldedness with the additional criterion that only cells with two fields in differently oriented alleys would not be included. This permissiveness was justified by the wide diversity of multi-field patterns seen in the data (*Figure 1—figure supplements 3–7*). Because it is not clear what drives repetition, and therefore what should count as repetition, a broad definition was employed to study the widest range of phenomena.

A metric termed the OAS was defined to test whether the apparent repeating patterns present within multi-fielded neurons were more common than the chance placement of fields would suggest. The OAS was a normalized measure that defined for each neuron how aligned the fields were compared to how aligned they possibly could be. First, the number of fields in vertical and horizontal alleys was defined. Next, the number of fields in the majority class was normalized by the total number of fields, yielding an alignment ratio. Next, this was compared to an a priori list of the possible alignment ratios for a cell with a given number of fields. Lastly, the ordinal position of the obtained alignment ratio was normalized by the number of possible alignment ratios for that number of fields.

As an example, consider a neuron with 6 fields, 4 of which are in vertical alleys. This is normalized to yield an alignment ratio of 4/6=0.67. Next, this ratio is compared to the list of possible alignment ratios for a neuron with 6 fields, which is [3/6=0.5, 4/6=0.67, 5/6=0.83, 6/6=1.0]. For an even number of fields the alignment ratio is minimal at 0.5 and maximal at 1.0. For an odd number of fields it is minimal at $((n+1)/2)/n$ and maximal at 1.0. The alignment scores cannot be lower than these bounds. In this example the alignment ratio of 0.67 is the second value in the list, out of four possibilities, so the final OAS value is 2/4=0.5. The OAS is minimal at 1/(number of alignment values) (here, 0.25) and maximal at 1.0.

To test for significance, the average OAS was computed across all multi-fielded neurons. Then, fields were randomly reassigned to neurons while maintaining the true total number of fields and a null OAS computed for each neuron. The average null OAS across multi-fielded neurons was computed and contributed one point to a null distribution. This was repeated 1000 times and the distribution of averaged OAS values was compared to the empirical average OAS with a one-sided statistical test at alpha = 0.05.

## Generalized linear model

To test whether individual fields encoded direction or time, GLMs were fit to the data. As mentioned elsewhere, if a field spanned multiple regions it was split into the portions on vertical or horizontal alleys separately. All GLM analysis was done in R (R version 4.0.3) using RStudio. GLMs were fit using the *glm* function from the *stats* package (*R Core Team, 2020*). The normalized firing rate was used as the dependent variable, with an offset of 1 added to make all values nonzero. Based on the observed skewness in the distribution of firing rates, the Gamma family was used with an inverse link function (this choice of family necessitated the +1 offset mentioned above). Current direction was represented as a binary variable. Models incorporating a time regressor represented time as the timestamp of the beginning of the field traversal. The resulting time series was fit with a natural spline with three knots using the splines package from R (*Bates and Venables, 2011*).

LRTs were used to test the significance of the model fits. This was done using the *lmtest* package in R (*Hothorn, 2021*). This test compares a base model to a more complex model containing a variable of interest. More complex models virtually always improve the model fit and the LRT tests whether this improvement is significant. The two variables tested here, in separate analyses, were current direction and time. It was suspected that in at least some cases the variables were correlated such that apparent effects of directional sampling could be due to time dynamics or, conversely, that apparent temporal dynamics were in fact due to changing biases in directional sampling over time. Therefore, when computing the LRT for the effect of current direction, the base model included a term for time. Conversely, when computing the LRT for the effect of time, the base model included a term for current direction.

The GLM had different sample size requirements than the Mann-Whitney test and therefore some neurons had sufficient sampling for the latter test but not the former. This is because while the Mann-Whitney test estimates one value for the whole population, the GLM includes two terms

whose regression weights are calculated along the length of the natural spline with three degrees of freedom. The GLM, therefore, has more parameters to estimate and this requires more data.

## Random forest classifier

A random forest classifier was used to decode direction from the ensemble firing rates (**Breiman, 2001**) using the scikit-learn package in Python (**Pedregosa et al., 2011**). For this analysis, place field boundaries were not used, as the activity of the entire recording ensemble was used regardless of where the cell fired. Behaviors when the animal turned around or was rewarded were removed as in other analyses. For each pass through each alley, a population vector was created. The ith entry of the vector was the normalized firing rate of the ith neuron. The normalized firing rate was computed as discussed above with the change that the session-wide activity in the alley was used. Each vector was associated with a directional label. The classifier used the population vector of firing rates as the input. Data were filtered by orientation such that direction was decoded from all horizontal alleys separately from the vertical alleys. The OOB score was used, as is standard for random forest classifiers. The OOB score is a measure of prediction accuracy based on testing trees on samples which they were not trained on (which occurs due to the bootstrap nature of the training procedure for each tree).

## Time series interpolation

The time series of each field's activity over time were correlated with one another. Because, in general, each time series had a different length, the vectors were interpolated to equate their sampling. The variability in time series length was a consequence of unequal sampling of each field due to the free foraging nature of the task. 1D piecewise cubic Hermite interpolating polynomial (PCHIP) interpolation was used because the fit was monotonic and would not overshoot the underlying signal (**Fritsch and Butland, 1984**). The implementation used scipy.interpolate.pchipinterpolator in Python (scipy, **Virtanen et al., 2020**).

## Linear decoder methods

A linear regression decoder was used to decode position from the ensemble firing rates. Spikes from each neuron were convolved with a Gaussian kernel (sigma = 25 camera frames) to create an instantaneous firing probability distribution. Data were also analyzed using a shorter convolution window (sigma = 8 frames) to test the effect of window size on the results; the pattern of results was not affected by convolution window size. Data were mean centered and normalized using sklearn.preprocessing.StandardScalar. X and Y positions were predicted from the population activity vector at each timepoint using sklearn.linear_model.LinearRegression. The $r^2$\_score function from scipy.metrics was used to estimate model performance as the amount of variance of the dependent variable (position) explained by the independent variable (population activity vector). Time epochs of alternating training (8000 points) and test (1000 points) windows were created. These windows were interspaced with buffers (500 points) so that the convolution process mentioned above did not cause data leakage. The decoder was trained on all training windows pooled together and tested on all testing windows pooled together. To estimate the variance in the decoder performance, the window boundaries were circularly rotated 1000 times, with shifts drawn from a uniform distribution. For the temporal decoding analysis (**Figure 6F and G**), the training windows were used to separately test on the rest of the windows. For the temporal decoding analysis the training windows were 10,000 points, the testing windows were 1000 points, and the buffers were 1000 points.

## Frequentist statistical tests

Mann-Whitney U tests were used as one method to test the directional firing of place fields. The mannwhitneyu function was used from the scipy.stats Python package. t-Tests were performed using scipy.stats.ttest_1samp. OLS regression was used to test the correlation within pairs of fields from the same corridor or different corridors (**Figure 5**). The statsmodels package in Python was used (**Seabold and Perktold, 2010**). A Fisher Z-test was used as an additional method to test the difference in correlations between the same corridor and different corridor field pairs (**Fisher, 1925**, pp. 161–168). The cocor package in R was used for this using the cocor.indep.groups function (**Diedenhofen and Musch, 2015**). A binomial test was used to compare different observed frequencies to those expected by chance. The function used was scipy.stats.binom_test with a one-sided (greater) test unless otherwise

indicated. Chi-square goodness of fit tests were used to test the relative frequencies of vertical versus horizontal repeating fields as well as repeating fields sharing or not sharing directional tuning. The scipy.stats.chisquare function was used. Pearson's chi-square test was used to test group differences between repeating and non-repeating fields for directional tuning and temporal dynamics (in separate tests). The scipy.stats.chi2_contingency function was used for this. Hartigan's dip test was used to test whether the trajectory response correlation distribution was not unimodal (*Figure 5G*). The 'diptest' package in R was used (*Maechler, 2021*). Data are shown as mean ± standard deviation, unless otherwise noted.

### Shuffling tests

For the same corridor/different corridor analysis, the labels of each pairwise comparison were pooled across the dataset. On each iteration, the labels were reassigned to the groups while preserving the sample sizes (same corridors = 54 pairs, different corridors = 155 pairs). Next, OLS regression was performed and the difference in $r^2$ values for $OLS_{same}$ and $OLS_{different}$ was stored as one entry in the null distribution. The 95th percentile of this distribution was compared to the real difference in $r^2$ values between the groups.

## Acknowledgements

This work was supported by R01 NS039456 awarded to JK, NSF GRFP #DGE1746891 awarded to WH, and T32 EY017203 which also supported WH. WH and JK contributed to the conception of the research question, task design, analysis, interpretation of the results, and writing the manuscript. WH contributed additionally to the data collection. R-YL contributed to the analysis and provided feedback on the manuscript; WS contributed to the analysis and provided feedback on the manuscript; MN contributed to the analysis and provided feedback on the manuscript. The authors would like to thank Geeta Rao, Arjuna Tillekeratne, Kimberly Nnah, and Kelly Wright for research support. The authors would like to thank Vyash Puliyadi, Ravikrishnan Jayakumar, Manu Madhav, and Cheng Wang for valuable feedback and assistance during the project.

## Additional information

### Funding

| Funder | Grant reference number | Author |
| --- | --- | --- |
| U.S. Public Health Service | R01 NS039456 | James J Knierim |
| U.S. Public Health Service | T32 EY017203 | William Hockeimer |
| National Science Foundation Graduate Research Fellowship Program | DGE1746891 | William Hockeimer |

The funders had no role in study design, data collection and interpretation, or the decision to submit the work for publication.

### Author contributions

William Hockeimer, Conceptualization, Data curation, Software, Formal analysis, Funding acquisition, Validation, Investigation, Visualization, Methodology, Writing – original draft, Writing – review and editing; Ruo-Yah Lai, Software, Formal analysis, Investigation, Methodology; Maanasa Natrajan, Software, Formal analysis, Visualization, Methodology; William Snider, Software, Formal analysis, Methodology; James J Knierim, Conceptualization, Software, Supervision, Funding acquisition, Methodology, Writing – original draft, Project administration, Writing – review and editing

### Author ORCIDs

William Hockeimer (i) http://orcid.org/0000-0002-2619-2528
James J Knierim (i) https://orcid.org/0000-0002-1796-2930

## Ethics

All animal procedures were approved by the Institutional Animal Care and Use Committee of Johns Hopkins University (Protocol number RA20A318) and complied with the Guide for the Care and Use of Laboratory Animals of the National Institutes of Health.

## Decision letter and Author response

Decision letter https://doi.org/10.7554/eLife.85599.sa1
Author response https://doi.org/10.7554/eLife.85599.sa2

---

# Additional files

### Supplementary files

Supplementary file 1. Alley traversals on which the animal was rewarded were excluded from all main text analyses as any CA1 response to the reward itself was an undesirable source of variability. However, major analyses were re-run with rewards included (i.e. all passes through the alley) to test whether this changed the results. In each of three major analyses, the inclusion of reward trials did not qualitatively change the results compared to when these trials were excluded. It should be noted for the re-analysis of *Figure 3B*, the numerator decreased by a large amount despite little change to the overall pool of fields tested. The fact that including rewards did not result in many more fields being included can be explained by the fact that rewards were uniformly distributed across the track so their removal decreased the samples from all fields evenly, resulting in little change to the number of fields included for analysis. The fact that the numerator (i.e. directional fields) decreased could be explained by the fact that rewarded trials introduced variability; indeed this was the justification for their removal, and this variability might have reduced the statistical power of the Mann-Whitney test from *Figure 3C*.

MDAR checklist

### Data availability

The data used in this manuscript are available at: https://doi.org/10.7281/T15HMQD4. Analysis code is available on GitHub (copy archived at *Hockeimer and Lai, 2025*).

The following dataset was generated:

| Author(s) | Year | Dataset title | Dataset URL | Database and Identifier |
|---|---|---|---|---|
| William H, Ruo-Yah L, Maanasa N | 2024 | Data associated with the publication: Leveraging place field repetition to understand positional versus nonpositional inputs to hippocampal field CA1 | https://doi.org/10.7281/T15HMQD4 | JHU Data Repository, 10.7281/T15HMQD4 |

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
