## [Editor Report]

This is a fundamental work that convincingly reveals that place cells in the hippocampus that exhibit repeated firing fields incorporate information about non-positional variables in each firing field. The authors reveal that individual firing fields of a single place cell can exhibit tuning to different head orientations, suggesting hippocampal neurons are flexible in terms of how they incorporate non-positional inputs.

---

## [Decision Letter]

**Decision letter after peer review:**

Thank you for submitting your article "Leveraging place field repetition to understand positional versus nonpositional inputs to hippocampal field CA1" for consideration by *eLife*. Your article has been reviewed by 2 peer reviewers, and the evaluation has been overseen by a Reviewing Editor and Laura Colgin as the Senior Editor. The following individual involved in the review of your submission has agreed to reveal their identity: André A Fenton (Reviewer #1).

Essential Revisions (for the authors):

1) Consider improving the clarity of the manuscript (R1 comments 2, 4, 5, 6, 11, 13, 14, 16, 17, 18, 21, 22), provide better justification for the use of the hybrid track (R1 comment 1) and include alternative interpretations of your results (R1 comments, 7, 12, 15, 19, 20).

2) Both reviewers noted that additional visualization and quantification of behavior would improve the manuscript. This could include examples of raw trajectories the animals took through the maze, controlling for head direction sampling to ensure that low levels of directional sampling are not influencing the results (R2 comment 3), and better quantification of whether or not the behavior was random foraging or more stereotyped.

3) R1 raised a point regarding theta cells or neurons that fired over larger portions of the box that should be addressed. Were these cells analyzed? Do they show repeating spatial firing? Could their concurrent activity be predicted or predict the activity of the place cells that were studied? Such analyses would go some ways to asking whether there is a global positional or non-positional signal.

4) Please clarify the process used to include/exclude cells. Why would cells like R886D2 TT5\Cl^-^maze1.1 be excluded from analysis and cells like R859D2 TT6\Cl^-^maze1.4 be included? They look to have similar features. There are many other examples like this.

5) Consider R2's suggestion to remove analyses of time, as it would strengthen the manuscript.

6) The GLM modeling results would be well served by comparison to some form of randomization. Random rotation of the time series for spiking relative to location and direction mapping data could be a useful approach. Here, the issue is to give some level of comparison for the likelihood ratio tests.

7) Do the authors have any data to address the question of whether repeating fields reflect similarity in view from each such location? Perhaps sessions where a full set of visual cues distinguishing walls of the environment were used? R2 has the concern that some will view the repeating fields as reflecting confusion despite the evidence for rate remapping across repeats.

*Reviewer #1 (Recommendations for the authors):*

What did the firing rate maps of theta cells look like? The examples in Figure S3 could be described and analyzed. Do they show repeating spatial firing? Could their concurrent activity be predicted or predict the activity of the place cells that were studied? Such analyses would go some way to asking whether there is a global positional or non-positional signal.

Why would cells like R886D2 TT5\Cl^-^maze1.1 be excluded from analysis and cells like R859D2 TT6\Cl^-^maze1.4 be included? They look to have similar features. There are many other examples like this.

*Reviewer #2 (Recommendations for the authors):*

Some suggestions for improvements are below:

1) The manuscript would be well served by removing the analyses and considerations of time in the episode. Representational drift cannot be dissociated from noise in the representation, from uncharted differences in the animal's experience of the environment across time (e.g., no. of visits to a location or differences in angles of approach), nor from developing satiation. The lack of demonstrated reproducibility in the time-based drift makes the finding and its interpretation highly questionable even taking into account prior work with similar claims and interpretations.

2) While the task is described as free-foraging, the authors should demonstrate that free or random-like foraging behavior was actually accomplished. It seems quite possible that a spontaneous regularity in search behavior could have emerged. It might also be interesting to compare volatility in speed and turning with open-field environments. Is there open-field data for these animals?

3) To what extent might the results concerning the directionality of in-field firing be influenced by an uneven sampling of directions across locations? Can the authors show directional tuning across more restricted (not 360-degree) ranges of directions for which robust sampling was available? Uneven sampling might even form a theoretical explanation for the emergence of the repeating fields in environments where path-running behavior repeats (e.g., Nitz 2011)

4) The GLM modeling results would be well served by comparison to some form of randomization. Random rotation of the time series for spiking relative to location and direction mapping data could be a useful approach. Here, the issue is to give some level of comparison for the likelihood ratio tests.

5) Decoder performance would be best described as distance error (in cms).

6) Do the authors have any data to address the question of whether repeating fields reflect similarity in the view from each such location? Perhaps sessions where a full set of visual cues distinguishing walls of the environment were used? My concern is that some will view the repeating fields as reflecting confusion despite the evidence for rate remapping across repeats.

---

## [Author Response]

Essential Revisions (for the authors):1) Consider improving the clarity of the manuscript (R1 comments 2, 4, 5, 6, 11, 13, 14, 16, 17, 18, 21, 22), provide better justification for the use of the hybrid track (R1 comment 1) and include alternative interpretations of your results (R1 comments, 7, 12, 15, 19, 20).

We provide the following clarifications based on the indicated comments and, where noted, have altered the text of the manuscript. We reply to the points in the order listed by the editor above:

R1 Comment 2: We use the terms ‘positional’ and ‘nonpositional’ to reflect functional inputs to CA1 that we identify through statistical analysis of CA1 place field firing rates. We fully agree that the larger hippocampal system, including the entorhinal cortex, may generate positional information rather than receiving it. This process clearly occurs in afferents to the hippocampus proper, in the form of grid cell and other spatial tuning properties of the MEC, so we can accurately refer to neural inputs to CA1 as carrying positional information. The main conclusion of the manuscript is that these inputs can be inferred due to the unique firing properties of repeating CA1 neurons. We show in Figure 2 that the fields of these neurons preferentially align to a common orientation which demonstrates, in our view, a positional input instructing the neuron at which type of alley to fire (horizontal vs. vertical). We show in Figure 5 that the directional tuning of these fields is not shared – neurons do not tend to prefer a given direction across all of their fields. To us, this demonstrates a non-positional (i.e., direction does not specify location) input that, further, is a field-level property, not a cell-level property. To make this point more clear we have added language on page 25 of the manuscript. We also replaced the word “study” with “infer” to address the reviewer’s concern that we are not actively manipulating these cues, but rather the design of the apparatus and the animal’s behavior allow us to infer these properties.

R1 Comment 4: We have changed the wording in the abstract to read: “thereby *offering a potential mechanism to* increase the flexibility…”

R1 Comment 5: The framework we refer to is one created by the brain which is taken to be subjective. We have added the word “subjective” as requested by the reviewer. The rest of the reviewer’s comment is addressed below, where we explain that by “inputs” we are talking about *neural* inputs to CA1, not inputs about putative objective space that are external to the organism. To avoid confusion, we have replaced phrasing related to “inputs to CA1” to phrasing describing “information represented in CA1” in various places throughout the manuscript.

R1 Comment 6: We do not believe that the studies cited by the reviewer are in any way inconsistent with the present results or can undermine the analyses we present. This comment appears to derive from the reviewer’s misinterpretation that our phrasing “inputs to the hippocampus” referred to inputs from outside the animal reflecting an objective space, rather than our intention to refer to neural inputs immediately afferent to the hippocampus (e.g., MEC, LEC, perirhinal, postrhinal, etc.). We agree that neural correlates of subjective spatial frames can be one of these neural inputs, perhaps generated in an upstream region of the hippocampus and relayed to CA1, but our experiment does not address this question. We hope that the changes we made to the manuscript to clarify our meaning alleviate this concern. We have added some reference citations on p. 26.

R1 Comment 11: The term “spatially-gated” means the variable response of the CA1 cell is dependent (“gated”) by spatial location. We have changed the wording to “spatially modulated” to avoid confusion.

R1 Comment 13: The term “slight trend” was intended to refer to the small magnitude of the effect, not whether the result was statistically significant. However, we acknowledge that this was ambiguous, as the term “trend” is often used to refer to a result that approaches the α level of the statistical test. We have changed the wording from “slight trend” to “small tendency” to avoid this confusion.

R1 Comment 14: We have changed the wording to “geometry”.

R1 Comment 16: We completely agree with the reviewer’s statement that conjoint activity of population activity is necessary to decode these signals. Indeed, we already demonstrated in the original manuscript that both positional (Figure 4C) and nonpositional information (Figure 3D,E) can be decoded at the population level, as the reviewer stated we need to do. Nowhere do we state that the system works only at the level of single cells or single place fields. Nonetheless, we believe strongly that it is imperative to understand how the system works at BOTH the single cell/single field level and the population level, and our analyses incorporate both levels of analysis.

R1 Comment 17: We do not fully understand the reviewer’s comment, especially why the reviewer brings in “future traversals” in this comment. The paper cited (Navratilova et al., 2012) shows precisely that directionality emerges over time with experience (Figures5, 6 of that paper). All we mean to suggest is that other forms of differential experience based on differences in the types of trajectories made as an animal moves within or between corridors may have some effect on the development of common directionality.

R1 Comment 18: We have changed the wording to use *trajectory* and *direction* throughout.

R1 Comment 21: There was a great deal of heterogeneity in the temporal responses of repeating fields of a single neuron (see Figure 6A and page 27). A cross-correlation between repeating fields could potentially identify a time offset with which to maximize field correlation, but it would not give further insight into how those fields change over time (see page 27 for further discussion). We do not believe we had the statistical power to rigorously characterize the diversity of time responses we saw in the data given our sample sizes.

R1 Comment 22: The time scale relevant to the analysis of Figure 6 is on the order of many seconds to minutes, and there are a number of possible mechanisms that might promote the firing rates of all of the firing fields of a repeating cell to wax and wane together at this time scale. For example, changing levels of neuromodulators or hormones might cause a cell’s activity to be generally higher or lower regardless of where in the apparatus the rat was located. One might imagine that other effects, such as changes in levels of global inhibition or changes in synaptic efficacy that last for minutes, might cause a cell’s activity to be generally higher or lower regardless of where in the apparatus the rat was located. However, our results suggest that such mechanisms are not in play, and that instead each firing field’s activity rate can be modulated independently, suggesting a more complex interaction between the positional inputs that determine where the cell should fire and other inputs that determine the magnitude of the firing rate at any moment. We have clarified this on page 27.

R1 Comment 1: We have added wording on page 10 to better explain the relevance of our track design. In brief, the design contains elements that, in previous studies, have been associated with directional tuning (linear or circular track segments in which the animal makes 2 oppositely oriented directional trajectories that may promote the formation of (quasi-)independent maps for each direction) along with elements that have been associated with a lack of directional firing (2-dimensional trajectories with no enforced, extended linear trajectories of opposite directions to associate with distinct maps). We hypothesized that combining these design elements would better explain what causes CA1 neurons to be directional – the existence of a goal-directed path to follow, or the physical geometry that perhaps leads to traversals being more easily distinguishable (through, e.g. different spatial views).

R1 Comment 7: We refer to the ‘rate remapping hypothesis’ as a specific hypothesis offered by Leutgeb et al. (2005) that addresses the function of rate remapping. The rate remapping *phenomenon* is that under some conditions, place cells maintain a similar firing location across different conditions but change their average firing rates in that location. The rate remapping *hypothesis,* according to Leutgeb et al. (2005), is that such a phenomenon might be a coding mechanism for episodic memory, as different episodes in the same location may be encoded by these differential firing rates. We have clarified this on page 3. We agree that this is speculative, but we believe that this is a useful way to consider these data, as rate remapping has become an important and well-studied phenomenon in the place cell literature.

R1 Comment 15: Figure 6A shows representative firing rate time series from neurons with non-stationary fields. It can be seen from this plot that even on the order of 10-15 minutes there is temporal variability. It is possible that the time constant of variability is somehow learned by the system based on the animal’s learning of the length of the experiment, and had we only recorded 10-15 minutes (instead of looking at a 10-15 minute block of a 60 min session) the dynamics would be different. This is speculative and our design does not allow us to probe this question. Absent such potential scaling, it stands to reason that a longer time course of the experiment would show more evidence of representational drift, if the rate of such drift is constant, and longer sessions are useful to detect the prevalence of this drift.

R1 Comment 19: A shift in reference frames is another plausible explanation for those data (although we do not see how it is more parsimonious), and we have added wording to include that possibility on page 26.

R1 Comment 20: We agree this would be an interesting and informative analysis to do. However, we are unable to address this question with our current data because we lack enough simultaneously recorded neurons to test this with sufficient power. For example, Wikenheiser and Redish (2015) required a minimum of 25 simultaneously recorded neurons for their analyses. In Figure 2A, we show that we did not have that many simultaneously recorded repeating neurons in our data set, since they were a subset of all recorded place cells. However, we believe that the discussion point is still an interesting speculation, and we have decided to keep it in the paper.

2) Both reviewers noted that additional visualization and quantification of behavior would improve the manuscript. This could include examples of raw trajectories the animals took through the maze, controlling for head direction sampling to ensure that low levels of directional sampling are not influencing the results (R2 comment 3), and better quantification of whether or not the behavior was random foraging or more stereotyped.

We include a new supplementary figure (Supplemental Figure 3) that details the behavioral processing steps using an example day from the dataset. The major processing steps are shown, showing for the full session what data is kept and removed at each step, as well as a 50 second example window of behavior showing what a snippet of behavior looks like at each step.

The directional sampling was set such that only alleys with at least two passes in both directions (e.g. at least two passes North and two passes South through a vertical alley) were included in the analysis.

Supplementary Figure 2 shows occupancy histograms of each alley as well as ethograms of the trajectories (defined as the previous, current, and next direction through each alley pass). These data show that the rats on average displayed some stereotypy in their travel through the maze (which is also true of ‘random foraging” tasks in an open field, in that rats display stereotyped movement such as thigmotaxis and other nonrandom trajectories). We believe our thresholds ensure we are only including alleys with enough sampling for statistical analysis.

3) R1 raised a point regarding theta cells or neurons that fired over larger portions of the box that should be addressed. Were these cells analyzed? Do they show repeating spatial firing? Could their concurrent activity be predicted or predict the activity of the place cells that were studied? Such analyses would go some ways to asking whether there is a global positional or non-positional signal.

We have added a new supplementary figure (Supplemental Figure 5) showing the rate maps of all putative interneurons (mean rate >= 10^0.6^ spikes/s). Some of these neurons did show firing patterns that appeared to be repeating (e.g., R859 D1 TT11 cell 2 and R859 D2 TT14 cell 4). However, we lack the statistical power to test whether their joint activity is predictive of or predicted by place field activity as we have an average of 2 interneurons simultaneously recorded per day, and some days had no interneurons at all.

4) Please clarify the process used to include/exclude cells. Why would cells like R886D2 TT5\Cl^-^maze1.1 be excluded from analysis and cells like R859D2 TT6\Cl^-^maze1.4 be included? They look to have similar features. There are many other examples like this.

Cells were initially excluded from analysis based on the absence of firing as the rat was engaged in forward motion on the track. Cell R886 D2 TT5 cell 1 was excluded because its firing field occurred mostly off-track during exploratory ‘scanning’ behavior as the rat reared along the wall of the maze. By contrast, the firing of cell R859 D2 TT6 cell 4 occurs within the bounds of an alley.

We have updated supplementary figure 4 such that each excluded neuron labeled in red now has a numeric code next to it detailing why it was excluded.

1. Place field detection algorithm did not detect any fields (i.e. firing hotspots too small and/or low rate), N = 2 neurons

2. Manual exclusion: Most firing rate off-track, often ‘scanning’ firing, N = 11 neurons

3. Manual exclusion: Firing mostly occurred as a burst when animal was placed on or taken off the track, N = 1

4. Manual exclusion: Algorithm’s field identification and partition did not match experimenter’s visual inspection, N = 1

5. Manual exclusion: Algorithm detected fields but there was excessive out-of-field firing, N = 5

5) Consider R2's suggestion to remove analyses of time, as it would strengthen the manuscript.

We believe that the time analysis is important for two reasons. First, it addresses Reviewer 1’s concern that we are treating place fields as static entities. Instead, this figure demonstrates that, at least on the time scale of minutes, place fields are demonstrably not static, and it is important to analyze this.

Second, the question of temporal drift is an important one in the study of the hippocampus (and the brain in general), and as R1 pointed out most hippocampal studies do not record single experimental sets for this long. Further, no study to our knowledge has addressed temporal drift in the context of multi-stable fields. Therefore, we believe these data in Figure 6, though observational, are still important to include in the manuscript to provide readers an accurate depiction of the data analyzed in this paper.

6) The GLM modeling results would be well served by comparison to some form of randomization. Random rotation of the time series for spiking relative to location and direction mapping data could be a useful approach. Here, the issue is to give some level of comparison for the likelihood ratio tests.

A randomization test was done to corroborate the GLM likelihood ratio test results of direction (Supplementary Figure 10). First, category labels (e.g., direction “N” or “S” for a vertically oriented field, “E” or “W” for a horizontally oriented field) were shuffled within each field 1000 times, maintaining the same number of each type of label but dissociating their relationship to firing rate. Then, the GLM likelihood ratio testing was done as for the real data and a proportion of fields that passed (spuriously) the test was computed. The proportion of fields that (spuriously) pass this test constituted one data point out of 1000 in the null distribution. We compare the 95^th^ percentile of the shuffle distribution to the real data and for direction and find the real proportion was well outside the shuffle distribution.

An analogous randomization test was done to corroborate the GLM likelihood ratio test results of time (Supplementary Figure 11). Time labels (“first”, “second”, “third” of the 3 thirds of the session) were shuffled within each field 1000 times, maintaining the true number of each label. In each iteration, the GLM likelihood ratio was performed as for the real data. The proportion of fields that (spuriously) passed the test was recorded. The shuffle distribution was therefore made up of the proportion of fields with a fictive response within each iteration. We compare the 95^th^ percentile of the shuffle to the real data and find the real value exceeds it.

7) Do the authors have any data to address the question of whether repeating fields reflect similarity in view from each such location? Perhaps sessions where a full set of visual cues distinguishing walls of the environment were used? R2 has the concern that some will view the repeating fields as reflecting confusion despite the evidence for rate remapping across repeats.

We did not explicitly control for the spatial view at each location. However, we did provide a black curtain next to the eastern wall of the maze as a polarizing cue and, because the recording system was also present next to the north wall of the maze, we believe there were sufficient external landmarks to resolve confusion. It’s important to note that while the maze had 6’’ high walls, these cues were visible above the walls and, further, the animals often reared up along the wall to peer outside which would allow them to orient themselves (note the presence of extra-maze sampling along the walls in many rate maps – this reflects the exploratory ‘scanning’ phenomenon).

Additionally, previous work has addressed whether repetition reflects confusion about current location. Singer et al., 2010 trained rats on an M-maze to perform an odor-guided memory task along the arms of the M-maze. They found that the animals could learn the task – indicating they could differentiate between these arms – yet repetition persisted. We have also added an additional reference to this paper on page 18.